# Efficient Rematerialization for Deep Networks

**Ravi Kumar**
Google Research
Mountain View, CA 94043
ravi.k53@gmail.com

**Manish Purohit**
Google Research
Mountain View, CA 94043
mpurohit@google.com

**Zoya Svitkina**
Google Research
Mountain View, CA 94043
zoya@google.com

**Erik Vee**
Google Research
Mountain View, CA 94043
erikvee@google.com

**Joshua R. Wang**
Google Research
Mountain View, CA 94043
joshuawang@google.com

## Abstract

When training complex neural networks, memory usage can be an important bottleneck. The question of when to rematerialize, i.e., to recompute intermediate values rather than retaining them in memory, becomes critical to achieving the best time and space efficiency. In this work we consider the rematerialization problem and devise efficient algorithms that use structural characterizations of computation graphs—treewidth and pathwidth—to obtain provably efficient rematerialization schedules. Our experiments demonstrate the performance of these algorithms on many common deep learning models.

## 1 Introduction

The world of deep learning is moving toward bigger model architectures. The recent successes in speech, language understanding, vision, and others have repeatedly demonstrated that bigger and deeper models yield the best results for a task, thereby advancing the state of the art. In addition to the size, the models themselves and the methods to train them are becoming increasingly complex and intricate in terms of data dependencies, gradient propagation, optimization steps, etc. Specialized hardware such as GPUs and AI accelerators have been vastly influential in training these complex models. They are particularly helpful from a computational point of view, but are limited by memory capacity that falls short of the peak demands of training these large models. Since memory turns out to be a bottleneck, it becomes an issue of feasibility—can a given model be trained at all?

While the growing model complexity is the root cause of severe demands on memory, the actual schedule in which the computation is carried out also plays a critical role in determining peak memory requirements. To see why, it is helpful to view the computational steps in training these models as a directed acyclic graph (e.g., Figure 1) whose nodes represent operations and directed edges represent data dependencies. (In TensorFlow parlance, this is a *dataflow* graph.) Each node consumes a set of inputs from its incoming edges, does some computation, and outputs the result of this computation on its outgoing edges; it is assumed that both inputs and outputs of this computation are to be held in memory. The order in which the nodes are computed, i.e., the schedule, will determine

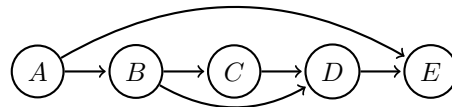

Figure 1: The schedule $\langle A, B, C, D, E \rangle$ needs four units of memory—while computing $D$, two units are needed for inputs to $D$, one for output from $D$, and one unit to keep the output of $A$ as an input to $E$. The schedule $\langle A, B, C, D, A, E \rangle$ needs three units of memory—at node $D$ the output of $A$ need not be retained in memory since it will be recomputed right after computing $D$.

the peak memory usage. Indeed, consider Figure 1, where the output of each node occupies one unit of memory. Computing the nodes in the order $\langle A, B, C, D, E \rangle$ would need four units of memory, whereas computing them in the order $\langle A, B, C, D, A, E \rangle$ would only need three units of memory (see caption of Figure 1). This latter order involves *rematerializing* the output of node $A$ instead of keeping it in memory. As this example illustrates, there can be a time-memory trade-off in our choice of schedule, where recalculating intermediate results can reduce what we store in memory. Judiciously choosing an appropriate schedule may make larger models feasible. In this paper we consider this rematerialization problem: given a computation graph as an input, construct a schedule, possibly rematerializing some nodes, that uses as little peak memory as possible[1].

When studied on computation graphs derived from training neural networks (i.e., graphs with forward computation and backward computations), rematerialization is often referred to as gradient checkpointing [10, 6, 8, 1, 15]. Of course, there are many other techniques to try to reduce memory usage, such as reusing memory regions [19] and trying to use both GPU and CPU memory [17, 16]. Rematerialization is a particularly nice approach because it only changes how the computation is done, but has no risk of changing the final result.

Compared to the gradient checkpointing line of work, we do not assume we have a forward/backward computation, but rather show how certain structural properties of the graph can be used to obtain a good solution. In particular, we identify *treewidth* of this graph as a key quantity that can be algorithmically exploited to yield a schedule with provable bounds on its length and peak memory usage. Informally, our main result is that there is a polynomial time algorithm that, given an $n$-node computation graph with treewidth $k$ and unit memory output at each node, constructs a schedule of length $O(kn^{\log k})$ and peak memory usage of $O(k \log n)$. This algorithm uses a tree decomposition of the computation graph, which yields balanced separators and offers a natural way to partition the computations into independent sub-computations while allowing us to bound the memory use through a charging argument. Note that while finding the optimal tree decomposition is computationally hard, there are efficient approximation algorithms and heuristics, which makes our algorithm efficient, practical, and easy to realize. We demonstrate its efficacy by applying it to training large networks including feedforward, residual, and transformer networks. In all these cases, our schedule yields significant savings in peak memory over baseline schedules, both with and without rematerialization.

We also design a different algorithm that produces schedules that are asymptotically more efficient. This algorithm relies on the path decomposition of the computation graph and is more intricate, with an involved analysis; but currently less practical. This result, however, hints at the intriguing possibility of another structural property of the graph that better captures its rematerialization potential.

## 2 Preliminaries

### 2.1 Computation Graphs and Schedules

The input to our algorithms is a *computation graph*. Each node in this graph represents an operation that takes as input zero or more tensors and produces a single tensor as an output (this assumption is for simplicity). Let $G = (V, E)$ be a directed acyclic computation graph. For $u, v \in V$, a directed edge $(u, v) \in E$ represents data dependency, meaning that the output of node $u$ is an input to node $v$. We are also given a *final* node $f \in V$ whose output tensor is required to be held in memory at the end of the computation. We assume, without loss of generality, that $f$ has out-degree zero (i.e. no other operations use the tensor produced by $f$) and all nodes in $G$ are needed to compute $f$. For any node $u \in V$, let $\text{in}(u)$ denote the immediate predecessors of $u$, i.e., $\text{in}(u) = \{u' \mid (u', u) \in E\}$. Let $n = |V|, m = |E|$, and $[n] = \{1, \ldots, n\}$. Throughout, $\log(\cdot)$ means $\log_2(\cdot)$.

A *schedule* for a computational graph $G = (V, E)$ is a sequence $\sigma = \sigma(G) = \langle u_1, \ldots, u_t \rangle$ of nodes in $V$ with the following properties: (i) the final node $f$ is represented in the schedule, and (ii) each node in the schedule occurs only after all of its predecessors, i.e., for each $j \in [t]$ and for each $u' \in \text{in}(u_j)$, there is some $j' < j$ such that $u_{j'} = u'$. Let $\text{prev}(u, j) = \max\{j' < j \mid u_{j'} = u\}$ be the most recent time $u$ occurs in the schedule before step $j$. Note that a node in $G$ can occur more than once in $\sigma(G)$ and this is precisely what enables the schedule length–memory usage trade-off.

---

[1] Of course, the real goal is to keep peak memory under the memory available and while minimizing the time to compute the schedule. Our results are easier to understand when viewed from a purely memory-minimization standpoint, but it is possible to stop our recursion early to obtain other trade-off points.

A schedule naturally implies time and memory bounds for computing $G$. Let $L(u)$ be the length of node $u$, representing the time required to execute the corresponding operation. The *length* of a schedule is given by $L(\sigma) = \sum_{i=1}^{t} L(u_i)$. Let $T_{\text{onepass}} = \sum_{u \in V} L(u)$ be the time required to execute every operation of the graph once. It lower bounds the length of any valid schedule.

The *peak memory* usage of the schedule, $M(\sigma)$, though intuitive, is a bit cumbersome to formalize. For $i \in [t]$, first define the set of tensors that need to be held in memory at step $i$ as

$$U_i = \{u_i\} \cup \text{in}(u_i) \cup \bigcup_{j > i} \{u' \in \text{in}(u_j) \mid \text{prev}(u', j) \leq i\}.$$

Let $s(u)$ denote the size of the tensor output by node $u$. Now, the memory of the schedule at step $i$ is $M(\sigma, i) = \sum_{u' \in U_i} s(u')$. Finally, $M(\sigma) = \max_{i=1}^{t} M(\sigma, i)$. The goal of an algorithm alg is to produce a schedule $\text{alg}(G)$ of $G$ that minimizes the peak memory. Let $M_{\text{in}} = \max_{u \in V} \{\sum_{u' \in \text{in}(u)} s(u')\}$ be the maximum input size needed to compute any tensor. Let $M_{\text{max}} = \max_{u \in V} s(u)$ be the maximum size of any tensor. Clearly, for any schedule $\sigma$, $M(\sigma) \geq \max\{M_{\text{in}}, M_{\text{max}}\}$.

## 2.2   Treewidth and Tree Decompositions

Treewidth is a well-studied graph parameter expressing how close an undirected graph $G = (V, E)$ is to a tree. Intuitively, if a problem is easy on trees, then one might hope that it remains easy on graphs of small treewidth. Formally, the treewidth of a graph is defined via the notion of tree decompositions. A *tree decomposition* of an undirected graph $G = (V, E)$ is a pair $(\mathcal{X}, T)$, where $\mathcal{X} \subseteq 2^V$ is a set of *bags*, with each bag a subset of the nodes, and $T$ is a tree on the bags $\mathcal{X}$. The bags and tree must satisfy the following three properties: (i) each node in $V$ is in some bag of $\mathcal{X}$, (ii) for each edge $(u, v) \in E$, both endpoints are together in some bag of $\mathcal{X}$, and (iii) for each node $v \in V$, the bags containing it (i.e., $\{X \in \mathcal{X} \mid v \in X\}$) form a connected subgraph of $T$.

Naturally, there are many tree decompositions of a particular graph $G$, including the trivial one that places all nodes into a single giant bag ($\mathcal{X} = \{V\}, T = \{\}$). We measure a tree decomposition by its *width*, which is the maximum bag size minus one: $\max_{X \in \mathcal{X}} |X| - 1$. The treewidth $\text{tw}(G)$ of $G$ is the minimum width of any tree decomposition. We refer to $|\mathcal{X}|$ as the *size* of the decomposition. Note that $\text{tw}(G)$ can range from 1 (a tree) to $n - 1$ (a clique).

We will use treewidth and tree decompositions of our directed computation graphs. When doing so, we are actually referring to the undirected graph obtained by forgetting the direction of every edge. It is known that series-parallel graphs have a treewidth of two and control-flow graphs of all programs written in C (without goto statements) have a treewidth of at most six [20]. We postulate that computation graphs of neural networks in the inference mode also have similarly low treewidth and that, given a computation graph $G$ for a neural network in the inference mode, the computation graph for training the network via backpropagation has treewidth at most twice as that of the original graph. Experimentally, we observe that computation graphs for training many common deep network architectures (ResNet, Transformer, and feedforward networks) have small treewidth (see Table 1).

Our results fall under the purview of fixed-parameter tractability, which studies the complexity of problems under particular parameters. Typically, we would hope to find an exact algorithm (computing the absolute memory-minimizing schedule) when treewidth is small. Unfortunately, this seems unlikely; such results typically come from Courcelle's theorem [7], which states that if a graph property can be expressed in second-order monadic logic, then it can be checked for in fixed-parameter tractable time relative to treewidth. Rematerialization is known to be PSPACE-complete [9]. If it were expressible in second-order monadic logic, then it would lie in the polynomial hierarchy (PH) and then PSPACE would collapse to PH. Hence, we must settle for approximation algorithms.

## 3   Efficient Rematerialization via Tree Decomposition

Our main algorithm uses a tree decomposition of the computation graph for a divide-and-conquer approach. The tree decomposition of a graph allows us to find balanced separators of small size. Additionally, the connectivity property of the tree decomposition guarantees that the nodes in the different components can be computed independently of each other except for interactions via the separator. Using these ideas, we recursively compute a memory-efficient schedule.

First, we consider the size of a tree decomposition and argue that it can be bounded.

**Lemma 1.** *Given an undirected graph $G = (V, E)$ and its tree decomposition $(\mathcal{X}, T)$ of width $k$, we can find another tree decomposition $(\mathcal{X}', T')$ of width $k$ and size at most $n$ in $O(|\mathcal{X}| \cdot (k + \alpha(|\mathcal{X}|)))$ time, where $\alpha(\cdot)$ is the inverse Ackermann function.*

*Proof.* The idea is to post-process $(\mathcal{X}, T)$ by repeatedly merging every adjacent pair of bags for which one bag is a subset of the other. This can be done with a single pass over all edges in $T$, since any adjacent pair of bags which cannot be merged at any time can never be merged in the future. For the sake of contradiction, imagine that bags $X_1$ and $X_2$ could not be merged due to some node $v \in X_1, v \notin X_2$. This problematic node will always be in $X_1$ since merging two bags only results in the addition of nodes to a bag. At the same time, it can never get added to $X_2$ because all bags that contain $v$ are connected and hence $X_1$ is the only bag in the neighborhood of $X_2$ that contains $v$.

We can keep track of these merges using a standard Union-Find data structure on the $|\mathcal{X}|$ bags, which costs $O(\alpha(|\mathcal{X}|))$ time per operation. We perform at most $|\mathcal{X}|$ merges, which cost a total of $O(|\mathcal{X}| \cdot \alpha(|\mathcal{X}|))$ time. To check whether one bag is a subset of another, we can put the larger bag in a hash set and perform $k + 1$ membership checks. Hence we can perform all these checks in $O(|\mathcal{X}| \cdot k)$ time. Hence the overall time is the claimed $O(|\mathcal{X}| \cdot (k + \alpha(|\mathcal{X}|)))$.

We can see why this post-processing procedure works by taking the resulting tree decomposition $(\mathcal{X}', T')$ and rooting it at an arbitrary bag. Each non-root bag must contain a node not found in its parent bag because otherwise the bag should have been merged with its parent bag. Since the set of bags containing a node is connected, this assigns a unique node $v \in V$ to every non-root bag. Furthermore, the root cannot be empty since then it would have been merged, and its nodes cannot be assigned to any other bag due to the same property. Hence we can assign it one of these nodes. Since we have assigned each bag a unique node $v \in V$, there can be at most $n$ bags. $\qquad\square$

A classic result shows that a tree always has a balanced node separator.

**Theorem 2** (Jordan [14])**.** *Any tree on $n$ nodes has a node whose removal disconnects the tree into components of size at most $n/2$.*

Applying Jordan's theorem on the tree decomposition $(\mathcal{X}, T)$ directly yields the following lemma.

**Lemma 3** (Balanced Separator)**.** *Given a tree decomposition $(\mathcal{X}, T)$, we can find, in time $O(|\mathcal{X}|)$, a bag $X^\star \in \mathcal{X}$ such that each connected component of $(\mathcal{X}, T) \setminus \{X^\star\}$ contains at most $|\mathcal{X}|/2$ bags.*

Our divide-and-conquer approach chooses a *balanced separator* $X^\star$ of the tree decomposition so that removing it results in subtrees with at most $|\mathcal{X}|/2$ bags each. Combining with Lemma 1, this guarantees that there are at most $\log n$ levels of recursion. Finding such a bag is a standard technique.

With these two ideas, we present Algorithm 1, which is a recursive function that schedules a subset $V'$ of nodes with a requirement that the schedule contains all nodes in a specified subset $S$. It breaks the graph using the balanced separator, and schedules the predecessors of a node $v$ in each of the resulting components before scheduling $v$ itself. The produced schedule includes annotations about which tensors to keep in memory or to remove, which is just for ease of analysis, as in practice memory usage can be inferred from a schedule of operations. Initially, the function is called with arguments $(G, V, (\mathcal{X}, T), \{f\})$, where $f \in V$ is the final node.

**Lemma 4.** *Algorithm 1 produces a valid rematerialization schedule.*

*Proof.* The base case of the recursion is when there is a single bag in the tree decomposition, in which case we make no recursive calls and simply compute the desired outputs in some topological order. Inductively, we assume that the algorithm works correctly on tree decompositions with less than $b$ bags, and show that it also works when there are $b$ bags.

The reasoning centers around what happens when we remove the balanced separator $X^\star$ from the tree decomposition. Since the bags containing any node $v \in V$ form a connected component, if $v$ is in two or more components of $\mathcal{C}$, it must also be in the separator $X^\star$. Hence this separator partitions our graph: for each subgraph $(\mathcal{X}', T') \in \mathcal{C}$, we can define the nodes in it to be $V' := \left( \bigcup_{X \in \mathcal{X}'} X \right)$ and we know that these $V'$ together with $X^\star$ form a partition of $V$. Furthermore, by the definition of tree decomposition, we know that each edge must be present in some bag, so the only edges involving some $V'$ go to other nodes in the same $V'$ or to $X^\star$.

---

**Algorithm 1:** Efficient Rematerialization via Tree Decomposition.

> **Function:** TWRemat$(G, V', (\mathcal{X}, T), S)$:
>
> > **Data:** $G = (V, E)$ a computation graph, $V' \subseteq V$ a subset of nodes to restrict to, $(\mathcal{X}, T)$ a tree decomposition of $G$ restricted to $V'$, $S \subseteq V'$ a subset of nodes to compute.
> > **Result:** An annotated schedule consisting of nodes in $V'$ that contains all nodes in $S$.
> >
> > **if** *this is the top level recursive call* **then**
> > > Shrink the size of the tree decomposition to at most $n$ bags using Lemma 1;
> >
> > Find a balanced separator (bag) $X^\star \in \mathcal{X}$ using Lemma 3;
> > Make a copy of $(\mathcal{X}, T)$, removing bag $X^\star$ and removing nodes of $X^\star$ from every other bag. Let $\mathcal{C}$ be the set of connected components that result (each a tree decomposition $(\mathcal{X}', T')$);
> > Initialize $schedule = \langle\rangle$;
> > **for** *node $v \in X^\star$ in any topological order (according to $G$)* **do**
> > > **for** *connected component $(\mathcal{X}', T') \in \mathcal{C}$* **do**
> > > > Let $S' = \text{in}(v) \cap \left(\bigcup_{X \in \mathcal{X}'} X\right)$ and $V'' = V' \cap \left(\bigcup_{X \in \mathcal{X}'} X\right)$;
> > > > Extend $schedule$ with TWRemat $(G, V'', (\mathcal{X}', T'), S')$ to compute the inputs of $v$ in this component;
> > > > Add annotation to $schedule$ to keep $S'$ in memory;
> > >
> > > Add $v$ to $schedule$, keeping it in memory, and freeing all of its inputs not in $X^\star$;
> >
> > **for** *connected component $(\mathcal{X}', T') \in \mathcal{C}$* **do**
> > > Let $S' = (S \setminus X^\star) \cap \left(\bigcup_{X \in \mathcal{X}'} X\right)$ and $V'' = V' \cap \left(\bigcup_{X \in \mathcal{X}'} X\right)$;
> > > Extend $schedule$ with TWRemat $(G, V'', (\mathcal{X}', T'), S')$ to compute the remaining outputs in this subgraph;
> > > Add annotation to $schedule$ to keep $S'$ in memory;
> >
> > Add annotation to $schedule$ to free the unneeded balanced separator nodes $X^\star \setminus S$;
> > **return** $schedule$;

---

We claim that whenever a recursive call to TWRemat is made (with arguments $V''$ and $S'$), all predecessors of $S'$ which are not in $V''$ are already in memory of the caller's schedule. Consider some node $u \in S'$ and its predecessor $u' \notin V''$. It must be that $u' \in X^\star$ by the preceding discussion that an edge involving $u \in V''$ can only go to $V''$ or to $X^\star$. Suppose that the recursive call is made from the nested for loops in which the outer loop is processing a node $v \in X^\star$. Since $u'$ is a predecessor of $u$ and $u$ is a predecessor of $v$ (which we know from $u \in S'$), $u'$ must come before $v$ in a topological order of $G$. Thus, it has already been scheduled in a previous iteration of the outer for loop. If the recursive call is made from the other for loop, then all nodes of $X^\star$ are scheduled and in memory by that time.

We conclude that the precedence constraints are respected by the schedule—with respect to nodes in $V''$ by induction, and with respect to nodes in $X^\star$ by the above discussion. Furthermore, all nodes of $S$ are scheduled in the later loop. $\qquad\square$

**Theorem 5.** *Given a computation graph $G = (V, E)$, its tree decomposition $(\mathcal{X}, T)$ of width at most $k$, and $S \subseteq V$ a subset of nodes to compute, Algorithm 1 runs in time $O(|\mathcal{X}| \cdot (k + \alpha(|\mathcal{X}|)) + kn \log n + kn^{1 + \log(k+2)})$ and computes a rematerialization schedule of length $O(T_{\text{onepass}} \cdot kn^{\log(k+2)})$ that requires $O((M_{\text{in}} + kM_{\text{max}}) \log n)$ memory.*

*Proof.* We begin with the running time. We pay an upfront cost of $O(|\mathcal{X}| \cdot (k + \alpha(|\mathcal{X}|)))$ to invoke Lemma 1. We pay a total time of $O(n \log n)$ to invoke Lemma 3, since (i) each invocation requires linear time and (ii) we recurse into subcalls that partition the tree decomposition into pieces that are at most half the current size. Note that we will need to memoize these balanced separators to avoid recomputing them over and over. As a result we have $O(\log n)$ levels of recursion and over all subcalls in a level we do $O(n)$ work. The processing of tree decompositions (removing a bag, removing the nodes of a bag from other bags) can be done in $O(kn)$ time and follows the same recursion as finding balanced separators (i.e. subcalls partition the tree decomposition and have at most $O(\log n)$ depth), for a total of $O(kn \log n)$ work. Finally, the output is $O(kn^{1 + \log(k+2)})$ in size (see the schedule length analysis), and we spend linear time to compute it.

Next, we check the schedule length. At each level, we make a recursive call to a particular subgraph $(|\mathcal{X}^*| + 1) \leq (k + 2)$ times, so we wind up amplifying the total work by a factor of at most $k + 2$ at each recursive level (except for the final recursive level, where we make no recursive calls). Carefully counting, we need at most $\lceil \log n \rceil + 1$ levels of recursion so we have amplified the computation time by $O((k + 2)^{\lceil \log n \rceil})$. Since $a^{\log b} = b^{\log a}$, this is an amplification of $O(kn^{\log(k+2)})$. In other words, we make at most $O(kn^{\log(k+2)})$ copies of any operation, so this takes at most $O(T_{\text{onepass}} kn^{\log(k+2)})$ time as claimed.

Finally, we check the memory needed by the schedule. Consider a particular segment of the schedule and the `TWRemat` function call that added it. The content of memory at this place in the schedule can be charged to the active function calls at that point of execution as follows: we charge to a recursive level everything that it annotated to keep in memory except its outputs, which are charged to its caller. The balanced separator requires $O(kM_{\text{max}})$ memory, one set of inputs to a balanced separator node requires $O(M_{\text{in}})$ memory (but since we free these we only need to hold one set of inputs). Since there are $O(\log n)$ levels of recursion, this results in a total memory of $O((M_{\text{in}} + kM_{\text{max}}) \log n)$. $\qquad\square$

What remains is to compute a tree decomposition efficiently. Our corollary utilizes an approximation algorithm that runs in $n \cdot 2^{O(\text{tw}(G))}$ time and computes a decomposition of width at most $(5\text{tw}(G)+4)$ and size $O(n)$ [2]. Our actual implementation uses a minimum fill-in heuristic [3], which yields good tree decompositions.

**Corollary 6.** *Given a computation graph $G = (V, E)$, there is an algorithm that runs in $2^{O(\text{tw}(G))}n + O(n \cdot (\text{tw}(G) + \alpha(n)) + \text{tw}(G)n \log n)$ time and computes a rematerialization that requires computation time $O(T_{\text{onepass}}\text{tw}(G)n^{\log_2(5\text{tw}(G)+6)})$ and memory $O((M_{\text{in}} + \text{tw}(G)M_{\text{max}}) \log n)$.*

## 4  Experiments

We experimentally evaluate the performance of our rematerialization algorithm on computational graphs for training commonly used deep neural networks. We remark that the memory optimizations proposed in this paper ensure that the computational graph is faithfully executed; this ensures that the gradients obtained at each train step are exactly equivalent to those obtained without any optimization, and hence do not affect convergence. We measure the theoretical peak memory usage of a schedule via an optimal static memory allocation plan. Since the primary purpose of these experiments is to evaluate the effect of rematerialization on memory usage, we do not consider other heuristic memory optimizations such as in-place operations, operation fusion, and buffer aliasing. Finally, we also measure the length of the schedule obtained by the different algorithms. For simplicity, in these experiments, we assume that each operation takes unit cost.

**Algorithms.** We compare the performance of the following three algorithms.

(i) `NoRemat`: Schedules all operations in a topological sort without any rematerialization.

(ii) `GreedyRemat`: This is an implementation of a greedy heuristic for rematerialization used by XLA[2] that works as follows. Starting with a topological sort of all operations, it processes each operation sequentially. At each stage, if the current memory usage is over a specified memory limit, the algorithm attempts to rematerialize an already scheduled operation. In particular, the operation whose rematerialization maximizes the amount of reduction in memory usage is chosen greedily at each step. If the memory usage cannot be reduced, the algorithm moves on to the next operation.

(iii) `TWRemat`: This is an implementation of Algorithm 1 that uses a tree decomposition; we use the minimum fill-in heuristic [3] to find the tree decomposition.

**Models and Setup.** We evaluate all algorithms on different families of widely used deep networks.

(i) *Deep Residual Networks (ResNet):* We first consider deep residual networks (ResNet) [13] as an example of convolutional networks for image classification. We use the official implementation of the ResNet model for the ImageNet task in TensorFlow[3]. We use different configurations to measure the effect of network depth (number of convolutional layers) on memory requirements of schedules obtained by the algorithms.

(ii) *Feed forward networks (FFN):* We consider a simple feed-forward neural network to illustrate the trends in peak memory usage of the schedules obtained by the different algorithms as a function of the network depth. For this experiment, we setup a simple feed-forward network with ReLU activations (number of hidden layers is varied) and randomly generated inputs and outputs. We use mean squared error loss and train using standard gradient descent.

| Model | $n$ | $m$ | tw |
|---|---|---|---|
| ResNet200 | 17,705 | 27,312 | 11 |
| FFN (100 layers) | 3,217 | 4,447 | 6 |
| Transformer Base | 15,842 | 21,771 | 18 |

Table 1: Computation graph statistics.

(iii) *Transformer:* We also evaluate the memory savings obtained by our rematerialization algorithms for training the transformer [21] network. Again, we use the official implementation of Transformer in TensorFlow[4] with all hyperparameters set to recommended defaults.

Table 1 gives summary statistics for representative models from each family. Crucially, we observe that even the largest graphs have tree decompositions with small width.

## 4.1 Effect on Peak Memory Usage

We first demonstrate the effect of the depth of the network on the peak memory usage required for training the network. Figure 2 compares the performance of the three algorithms on the ResNet and Feed-forward models described above. As expected, we observe that the peak memory usage of `NoRemat` that does not perform any rematerialization increases linearly with the number of layers on both model families. The `GreedyRemat` algorithm yields modest improvements ($\approx$ 2x) in memory usage for the ResNet models but still shows a linear growth with number of layers. We observe that `GreedyRemat` yields very little memory savings on the feed forward network. On the other hand, the `TWRemat` algorithm consistently gives memory savings on both the model families (up to 10x) and the growth in peak memory usage is distinctly sublinear.

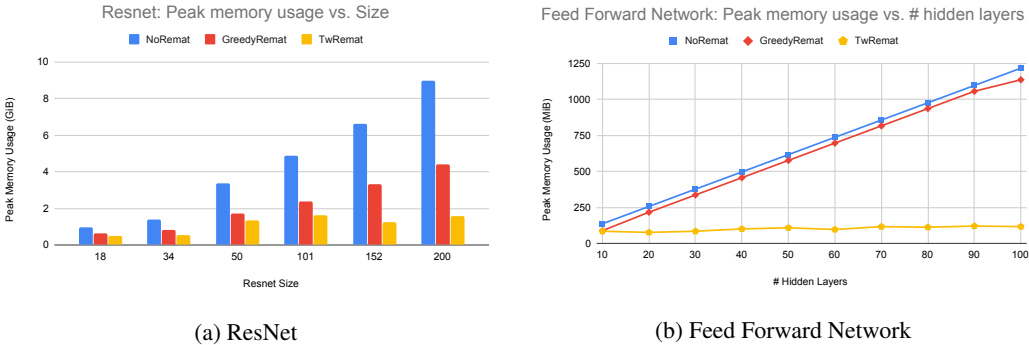

(a) ResNet          (b) Feed Forward Network

Figure 2: Peak memory usage vs. model depth.

Table 2 shows the memory usage and relative lengths of the schedules obtained by the three algorithms on two configurations of the transformer network. The `TWRemat` algorithm yields a 3.48x and 4.59x reduction in peak memory usage respectively, albeit at a cost of up to 10.6x in the schedule length.

| | NoRemat | GreedyRemat | | TWRemat | |
|---|---|---|---|---|---|
| | Mem. (GiB) | Mem. (GiB) | Rel. Len. | Mem. (GiB) | Rel. Len. |
| Transformer Base | 3.97 | 2.92 | 1.21 | 1.14 | 10.61 |
| Transformer Big | 13.25 | 10.12 | 1.27 | 2.89 | 10.64 |

Table 2: Transformer: Peak memory usage and relative schedule lengths.

## 4.2 Effect on Schedule Length

Our algorithms are specifically designed to minimize peak memory consumption at the expense of additional computation. Figure 3 illustrates the increase in the schedule length relative to `NoRemat`.

We observe that `GreedyRemat` consistently yields schedules that are only marginally longer than the corresponding schedules of `NoRemat`. On the other hand, the schedules obtained via `TWRemat` are around 3x-4x longer. Despite the longer schedules, we expect the schedules produced by `TWRemat` to be beneficial in practice as the reduced memory usage allows the use of specialized hardware accelerators.

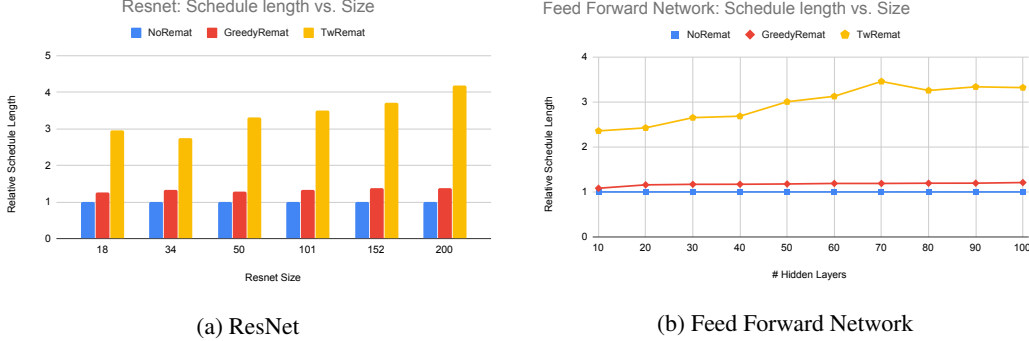

|         |          |
|:-------:|:--------:|
| (a) ResNet | (b) Feed Forward Network |

Figure 3: Schedule length vs. model depth.

## 4.3 Trading-off Memory Usage for Schedule Length

Algorithm 1 (`TWRemat`) uses the tree decomposition to find a balanced separator that breaks up the tree decomposition into smaller subtrees, and then recursively computes schedules to compute the required nodes in these subtrees. We observe that we can obtain a trade-off between memory usage and schedule length by preemptively stopping the recursion when the tree decomposition has few bags remaining. For any integer $k$, let `TWRemat` $(k)$ be a variant of Algorithm 1 that stops the recursion when the tree decomposition has fewer than $k$ bags. In the base case, we schedule the required nodes in an arbitrary topological order. In this notation, our `TWRemat` algorithm can be written as `TWRemat` (1). Indeed, by varying the recursion limit from $k = 1$ to $k = n$, we can interpolate between the `TWRemat` and `NoRemat` algorithms. Figure 4 shows the memory usage vs. schedule length trade-off obtained for the ResNet200 model.

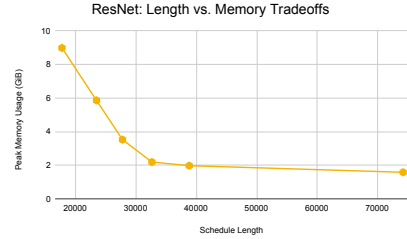

Figure 4: Resnet200: Mem. vs length.

## 5 Stronger Guarantees via Path Decomposition

Related to treewidth of a graph is the notion of *pathwidth*, which is defined as the minimum width of any *path decomposition*, where a path decomposition is a tree decomposition $(\mathcal{X}, T)$ under the additional constraint that $T$ must be a path. We can order the bags according to the path and instead use the tuple $\mathcal{X} = (X_1, X_2, ..., X_{|\mathcal{X}|})$ to represent the path decomposition, where each $X_i \subseteq V$ is a bag and (path decomposition) edges run between $X_i$ and $X_{i+1}$. We denote the pathwidth of a graph $G$ by $\mathrm{pw}(G)$. Assuming that a computation graph has a small constant pathwidth allows us to design an algorithm for rematerialization that leverages the path decompositions to yield stronger theoretical guarantees than in Theorem 5. In this section, we sketch the primary ideas, deferring the full algorithm and analysis to the Supplementary Material.

We first show one can add a directed Hamiltonian path (i.e., a *spine*) to any graph $G$ so that the pathwidth of $G$ only increases by a factor of $\sim 2$. This allows us to prove certain structural properties of the path decomposition. Suppose the vertices of $G$ are ordered according to the spine, let $u_i \in V$ be the $i$th node, and let $last(X)$ denote the index of the last node in bag $X$. We show that if $X_c \in \mathcal{X}$ is a bag in the path decomposition that contains $u_n$, then for all $\ell < \ell' < c$, we have $last(X_\ell) \leq last(X_{\ell'}) \leq last(X_c)$ and for all $r > r' > c$, we have $last(X_r) \leq last(X_{r'}) \leq last(X_c)$.

Such a structural characterization allows a divide-and-conquer strategy that recurses on the *right* and *left* sides of the path decomposition. Unlike the tree decomposition algorithm where we argue that

the size of the tree decomposition reduces at each recursive call, the additional properties of the path decomposition allow us to argue that *both the size and width* of the decomposition decreases. The resulting algorithm yields a schedule that incurs a polylogarithmic increase in length (vs. polynomial blow up for the tree decomposition), but at the cost of polylogarithmic memory usage.

## 6  Related Work

Rematerialization has been considered in very limited settings for training deep networks. The work most relevant to ours is that of Chen et al. [6] and Gruslys et al. [11]. The former shows how to trade off memory and computation cost for simple chain-like networks. Their algorithm at a high level works by dividing a computation of length $n$ into $\sqrt{n}$ many sub-computations, storing the internal states for each sub-computation and at the $\sqrt{n}$ check points; a second pass is needed to complete the computations. By recursing on this idea, one can get an $O(n \log n)$-pass algorithm using memory $O(\log n)$ for chain-like computations. Gruslys et al. [11] consider backpropagation through time and propose a dynamic-programming based approach for achieving the best time-memory trade off; their algorithm is tailored to work on RNNs. It is unclear how to extend either of these algorithms to work for general computation graphs, which is the focus of our work. There are some practical heuristics for rematerialization used in open-source efforts such as XLA; in fact, we used it as one of our baselines (`GreedyRemat`). Other heuristics including in-place operations and register sharing memory optimizations have been used in practice [5]. We, on the other hand, offer a principled approach to these problems.

Tree decomposition has been suggested as a tool to achieve time-memory trade off in register allocation problems in compilers [18, 4]. A recent blog post[5] informally suggests using tree decomposition for memory saving in deep networks in the context of gradient checkpointing,[6] which implements [6]. As noted control flow graphs of structured programs have treewidth $\sim 6$ [20]. Here, we work with the data flow graph to obtain a memory-efficient schedule, which may have larger treewidth in general.

View materialization in databases is also somewhat related to rematerialization [12]. The goal there is to pre-compute materialized views in order to efficiently answer future queries. While this is also a computation-memory trade-off, the end goals are clearly different from our setting.

## 7  Conclusions

We consider the rematerialization problem in the context of memory-efficient training of deep networks and obtain efficient algorithms based on tree decomposition for finding a provably good schedule with rematerialization. Although our path decomposition based algorithm yields asymptotically better schedules, the schedule length and memory depend exponentially on the pathwidth. It will be very interesting to make this algorithm more practical. Identifying the precise structural parameter that characterizes rematerialization of a given graph is a tantalizing research question.

## Footnotes

[2] `www.tensorflow.org/xla`

[3] `github.com/tensorflow/models/blob/master/official/resnet/imagenet_main.py`

[4] `github.com/tensorflow/models/blob/master/official/transformer/transformer_main.py`

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
