[Supplementary Material]

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

[1] Of course, the real goal is to keep peak memory under the memory available and while minimizing the time to compute the schedule. Our results are easier to understand when viewed from a purely memory-minimization standpoint, but it is possible to stop our recursion early to obtain other trade-off points.

[2] `www.tensorflow.org/xla`

[3] `github.com/tensorflow/models/blob/master/official/resnet/imagenet_main.py`

[4] `github.com/tensorflow/models/blob/master/official/transformer/transformer_main.py`

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

# A  Stronger Guarantees via Path Decomposition

In this section, we describe an algorithm to find a schedule using polylogarithmic space and polylogarithmic blowup in schedule length for graphs having a low pathwidth. The main goal is to prove the following result.

**Theorem 7.** *Let $G = (V, E)$ be a computation graph with pathwidth $\mathrm{pw}(G)$. Then we can compute a schedule $\sigma$ for $G$ using memory $M(\sigma) = O((M_{\mathrm{in}} + M_{\max})4^{\mathrm{pw}(G)}(\log^{2\mathrm{pw}(G)+1} n))$ and length $L(\sigma) = 2^{O(\mathrm{pw}(G) \log \mathrm{pw}(G))} n (\log^{2\mathrm{pw}(G)+1} n)$.*

*In the special case that $G$ also has a directed path of length $n$ (i.e., a directed path touching every node), then we can compute a schedule $\sigma'$ for $G$ using memory $M(\sigma) = O((M_{\mathrm{in}} + M_{\max})2^{\mathrm{pw}(G)}(\log^{\mathrm{pw}(G)-1} n))$ and length $L(\sigma) = 2^{O(\mathrm{pw}(G) \log \mathrm{pw}(G))} n (\log^{\mathrm{pw}(G)-1} n)$.*

The proof is written across four subsections. In the first subsection, we explain how to do some preprocessing to establish a useful property that we will need for future subsections. Specifically, we show that for any graph $G$ with pathwidth $\mathrm{pw}(G)$, we can add edges to $G$ to create $G'$ such that $\mathrm{pw}(G') \leq 2\mathrm{pw}(G) + 3$ and further, $G'$ contains a directed path of length $n$. In this case, we say $G'$ has a *long spine*. Note that this is why we do even better on graphs that already come with a long spine; we can skip this preprocessing step.

The second subsection then leverages the long spine to prove useful structural properties of the path decomposition that will help us appropriately recurse. In particular, we explore ideas that will allow us to design an algorithm that reduces both the size (number of bags) and width (number of nodes in each bag) of the path decomposition as we recurse. For tree decompositions, we were only able to lower the number of bags; this is why we will be able to obtain improved bounds for path decompositions.

The third subsection helps us with the all important step of combining the schedules that work for two subproblems into a schedule for the current problem. To do so, we introduce the notion of "interleaved schedules" and prove several key properties about them that we will need for the algorithm.

Finally, the fourth subsection presents the algorithm and analyzes its correctness. It also bounds the length and memory usage for the resulting schedules.

## A.1 Pathwidth-Preserving Spine-Addition

The goal of this subsection is to prove that a spine can be added to a computation graph while controlling the width of its path decomposition. It is important that we choose the spine to add, since it is not hard to construct counterexamples where the addition of a poorly-chosen spine increases pathwidth by a $\mathrm{poly}(n)$ factor. As a reminder, we restate the theorem we wish to prove.

**Theorem 8.** *Suppose we have a directed acyclic graph $G$ on $n$ nodes and a path decomposition of $G$ with bag size $\mathrm{pw}(G) + 1$. Then there exists an algorithm that adds a spine to $G$ while maintaining a valid path decomposition. The resulting path decomposition has bag size at most $2\mathrm{pw}(G) + 3$.*

*Proof.* We begin with some helpful notation. Given a path decomposition $(\mathcal{X}, T)$ we can label its bags $X_1, \ldots, X_b$ in path order. Define $\textsc{Interval}(u) := \{i \in [b] \mid u \in X_i\}$. By the definition of path decomposition, this is always some contiguous interval $[\ell_u, r_u]$ since these bags form a subpath.

We will prove that Algorithm 2 has the desired properties. First, we will show that a spine is added to $G$. To do so, we first prove that all nodes are processed in some topological order. Clearly nodes are only processed in topological order, since we refuse to process any node before its in-neighbors. Hence it remains to show that all nodes are processed; which we do by contradiction. Fix an arbitrary topological order of $G$, and for the sake of contradiction suppose that $v$ is the first node in this order which is not processed by our algorithm. All of its in-neighbors must be processed, or it wouldn't be the first such node. Furthermore, there must be at least one such in-neighbor, $r$. When its last in-neighbor is processed, it had no unexplored in-neighbors and hence $v$ should have been processed. Hence, all nodes are processed.

We claim that when `ProcessNode(..., u)` is called, it adds a path to $G$ starting from $u$ and covering all nodes that appear in recursive subcalls to `ProcessNode`. This is because for the first recursive subcall, we already know there is a $u \to v_1$ edge and before subsequent recursive subcalls, we add an edge from the current end of the path to the next $v_i$. Since we already know `ProcessNode(..., r)` results in all nodes being processed, the path it adds must covers all nodes, i.e., must be a spine.

Now, we will show that we have properly updated the path decomposition and that no bag has more than $2\mathrm{pw}(G) + 3$ nodes. The former is by construction; we never add an edge without first ensuring that its endpoints share a bag. Regarding the latter, note that we begin with bags of size at most

---

**Algorithm 2:** Pathwidth-Preserving Spine Addition.

---

**Function:** PathwidthPreservingSpineAddition($G, (\mathcal{X}, T)$):

   **Data:** $G = (V, E)$ a DAG and $(\mathcal{X}, T)$ a path decomposition of $G$.
   **Result:** Adds a spine to $G$ and updates the path decomposition accordingly.

   Add a node $r$ to $G$, connect it to all other nodes, and add it to all bags $X_i \in \mathcal{X}$;
   Run ProcessNode($G, (\mathcal{X}, T), \{\}, r$);
   Remove node $r$ from $G$ and from all bags $X_i \in \mathcal{X}$;

**Function:** ProcessNode($G, (\mathcal{X}, T), S, u$):

   **Data:** $G = (V, E)$ a DAG, $(\mathcal{X}, T)$ a path decomposition of $G$, $S \subseteq V$ a subset of explored nodes, and $u$ a node of $G$.
   **Result:** Adds a path starting from $u$ and covering all recursively processed nodes; returns the final node $x$ of this path. Updates the path decomposition to handle this path and guarantees INTERVAL($x$) includes $r_u$.

   Add $u$ to the explored set $S$;
   Set the current final node $x \leftarrow u$;
   Let $v_1, v_2, \ldots, v_k$ be the out-neighbors of $u$ which have no unexplored in-neighbors of their own, ordered by increasing right endpoint $r_{v_i}$;
   **for** $i = 1, 2, \ldots, k$ **do**
      Update $x \leftarrow$ ProcessNode($G, (\mathcal{X}, T), S, v_i$);
      **if** $i < k$ **then**
         Extend the interval of $x$ to the right until it includes $\ell_{v_{i+1}}$;
         Add edge $x \rightarrow v_{i+1}$ to $G$;
      **else**
         Extend the interval of $x$ to the right until it includes $r_u$;

   **return** $x$;

---

$\mathrm{pw}(G) + 1$, we add $r$ to all of them for bags of size at most $\mathrm{pw}(G) + 2$, we call ProcessNode($\ldots, r$) which will double the bags to size at most $2\mathrm{pw}(G) + 4$, and finally removing $r$ from all bags results in bags of size at most $2\mathrm{pw}(G) + 3$ (our procedure only adds nodes to bags, so $r$ is still in all bags).

The tricky part is proving that ProcessNode($\ldots, r$) at most doubles the size of all bags; this is due to two observations. The first observation is that if we examine how nodes are added to bags inside ProcessNode($\ldots, u$) ignoring all recursive subcalls, at most one node gets added to any bag and only to bags already containing $u$. After $x \leftarrow$ ProcessNode($\ldots, v_i$) we know that the right endpoint $r_x$ is at least the right endpoint $r_{v_i}$. Hence during any iteration $i \in [k]$, at most one node gets added to any bag strictly to the right of $r_{v_i}$. But before iteration $i \in [k]$, nodes could only be added to bags as far right as $r_{v_i}$, since the left endpoint of an interval precedes the right endpoint of an interval and since $r_{v_1} < r_{v_2} < \cdots < r_{v_k}$. Nodes cannot be added to bags to the left of INTERVAL($u$) because $r_{v_1} \in$ INTERVAL($u$). Nodes cannot be added to bags to the right of INTERVAL($u$) because $\ell_{v_k} \in$ INTERVAL($u$).

The second observation is that the INTERVAL($u$) is only altered after ProcessNode($\ldots, u$) completes. Hence ProcessNode($\ldots, u$) can only add a node to each bag that $u$ was *originally in*. As a result, each bag can only gain a node for each node that was originally in it, so each bag at most doubles in size. This completes the proof. $\qquad\square$

## A.2 Structural Properties for Long-Spined Graphs

Let $G = (V, E)$ be a directed acyclic graph on $n$ nodes that has a long spine, i.e., there is a directed path of length $n$ through $G$. Throughout, let $\mathcal{X} = X_1, X_2, \ldots, X_b$ be a path decomposition for $G$, and let $p = \mathrm{pw}(G) + 1$ be the maximum bag size in $\mathcal{X}$.

Since we have a path through the entire graph, there is a full topological ordering forced on us. Label the path in order $v_1 \prec v_2 \prec \cdots \prec v_n$, so $v_1$ is the first node and $v_n$ is the last. It now makes sense to talk about the *last* element in a bag—it is the node $v_i$ with the largest index $i$. Formally, for any bag $X$, let last($X$) be the last node in $X$.

Path decompositions of DAGs with long spines have several useful properties which we will need. The first property states, roughly, that any path passing between two bags must also pass between any intermediate bag.

**Lemma 9** (Intermediate Value Theorem for Bags). *Define $G$ as above, and suppose we have three bags in our path decomposition, say $X_k, X_{k^*}$, and $X_{k'}$ with $k \leq k^* \leq k'$. Furthermore, suppose that $v_i \in X_k$ and $v_{i'} \in X_{k'}$ for some $i, i'$. Then there is an $i^* \in [\min(i, i'), \max(i, i')]$, such that $v_{i^*} \in X_{k^*}$.*

*Proof.* We induct on the difference between the nodes, $|i - i'|$. Our base cases are $|i - i'| = 0$ and $|i - i'| = 1$. The first base case $i = i'$ trivially follows from the definition of path decomposition (in particular, the bags that contain $i = i^* = i'$ form a contiguous interval). The second base case $|i - i'| = 1$ is a bit trickier to argue about. Our long spine implies an edge from $v_i$ to $v_{i'}$ (or vice versa). No matter which way this edge runs, by the definition of path decomposition there is some bag $X_\ell$ that contains both $v_i$ and $v_{i'}$. If this bag is to the right of our goal bag, i.e., $k^* \leq \ell$, then from the definition of path decomposition we can deduce that $i$ is in the goal bag (it is in $k \leq k^*$ and $\ell \geq k^*$). If this bag is to the left of our goal bag, i.e., $\ell \leq k^*$, then from the definition of path decomposition we can deduce that $i'$ is in the goal bag (it is in $\ell \leq k^*$ and $k' \geq k^*$). Hence in either case the we can find an $i^*$ as desired.

The inductive case is similar to the second base case. We will focus on the situation where $i < i'$ (the proof for the $i > i'$ case is analagous). The long spine implies an edge from $v_i$ to $v_{i+1}$ and hence there is some bag $X_\ell$ that contains both $v_i$ and $v_{i+1}$. If this bag is to the right of our goal bag, i.e., $k^* \leq \ell$, then we can apply our inductive hypothesis to the simpler problem $(k \leftarrow k, k^* \leftarrow k^*, k' \leftarrow \ell, i \leftarrow i, i' \leftarrow i+1)$. If it is to the left of our goal bag, i.e., $\ell \leq k^*$, then we can apply our inductive hypothesis to the simpler problem $(k \leftarrow \ell, k^* \leftarrow k^*, k' \leftarrow k', i \leftarrow i+1, i' \leftarrow i')$. Hence in either case we can find an $i^*$ as desired. This completes the proof. $\square$

Figure 5: The last$(\cdot)$ function is single-peaked. Notice that the peak may stretch over multiple bags (in this case, $X_3$ and $X_4$).

The next property is easier to understand with a picture; refer to Figure 5. Suppose we plot last$(X_i)$ on a graph with the $x$-axis representing bag indices and the $y$-axis representing node indices. Then the curve traced out by last$(\cdot)$ will have a single peak; it is nondecreasing before the peak and nonincreasing after. More formally, we have the following lemma.

**Lemma 10.** *Suppose the final node $v_n$ is in some bag $X_c$. Then for any $k, \ell$ such that $c \leq k \leq \ell$, last$(X_k) \geq$ last$(X_\ell)$. Additionally, for any $k', \ell'$ such that $\ell' \leq k' \leq c$, last$(X_{\ell'}) \leq$ last$(X_{k'})$*

*Proof.* We prove just the $c \leq k \leq \ell$ case (the proof of the $\ell' \leq k' \leq c$ case is analogous). Let $i = \text{last}(X_\ell)$. By Lemma 9, there is a $i^* \in [i, t]$ such that $v_{i^*} \in X_k$. This node is a lower bound on $\text{last}(X_k)$; we deduce that $\text{last}(X_k) \geq i^* \geq i = \text{last}(X_\ell)$. This completes the proof. □

This final property will be our main tool to bound the complexity of recursing. The main idea it tries to capture is that some particular ways to restrict our graph $G$ into a subgraph $G'$ result in stripping the last node from every bag and hence lower the width of the path decomposition by one.

**Lemma 11.** *Suppose that the final node $v_n$ is in some bag $X_c$. Then for any $k, \ell$ such that $c \leq k \leq \ell$, the following is true. Let $G'$ be $G$ restricted to nodes with indices in the range $[1, \text{last}(X_\ell) - 1]$ and that appear in bags from $X_k$ to $X_\ell$. Let $G''$ be $G'$ with edges $(u, v)$ added if both $u$ and $v$ are topologically before $\text{last}(X_\ell)$ and either (1) $u, v \in X_k$ or (2) $u, v \in X_\ell$. Note that $G''$ has a long spine, i.e., a directed path through $G''$ that touches every node of $G''$.*

*Then $G''$ has a path decomposition with bag size of $p - 1$. In fact, if we take the original path decomposition for $G$ from $X_k$ to $X_\ell$ and we remove all nodes not in $G''$, that is a valid path decomposition for $G''$ with bag size $p - 1$.*

*An analagous statement holds for bags on the "left side" ($\ell \leq k \leq c$).*

*Proof.* In general, given graph $H$ and subgraph $H'$, if we have a path decomposition for $H$, then when we remove all nodes not in $H'$, that is a valid path decomposition for $H'$: The edge property still holds, and the between-ness property still holds. (We can also remove all empty bags while maintaining the path decomposition properties.)

So let $\mathcal{X}$ be the path decomposition for $G$, and let $\mathcal{X}''$ be the same decomposition when restricted to nodes in $G''$. Since every edge we added to $G''$ must be in either $X_k$ or $X_\ell$ (even after restricting to nodes in $G''$), $\mathcal{X}''$ is a valid path decomposition for $G''$.

So we only need to show that for all $X \in \mathcal{X}''$, the size of $X$ is at most $p - 1$. But $X$ is to the left of $X_\ell$, so $\text{last}(X) \geq \text{last}(X_\ell)$. Hence, $X$ restricted to nodes in $G'$ does not contain $\text{last}(X)$. That is, it has size at most $p - 1$. This completes the proof. □

Adding edges to $G''$ may seem like a strange technical condition, but it is much more natural to think about it in the following way. Given a path decomposition for $G$, it induces an interval for each node in $G$, as described in Section A.1. This in turn induces an interval graph: we have an edge between $u$ and $v$ iff the intervals for $u$ and $v$ overlap. This interval graph is necessarily a supergraph of $G$. Throughout our algorithm, we actually operate on this supergraph. After all, in the worst case, the induced interval graph and $G$ are identical. (The one exception to this is that the actual memory used by the schedule is based on the true indegree of nodes rather than the expanded indegree.)

Note that in an interval graph, these edges that we would add already exist; i.e., $G'' = G'$.

## A.3 Key Properties of Interleaved Schedules

Our recursive algorithm will need to take in a subgraph (along with a path decomposition) and return a low-memory schedule. To properly use the schedules returned by our recursive subcalls, we need an understanding of how to interleave them together. We will go over the properties here; the matching proofs appear in Subsubsection A.3.2. Also, note that we will be assuming throughout this subsection that there is a complete topological ordering on the vertices under consideration.

**Definition 12.** *Suppose we have a set of nodes $W \subseteq V$. A schedule $\sigma = u_1 u_2 \ldots u_t$ is said to be valid for known set $W$ if for every $u_i \in \sigma$, either (1) $u_i \in W$, or each predecessor of $u_i$ either (2a) is in the known set $W$ or (2b) appears earlier in the schedule.*

We use $\circ$ to denote concatenation; for example $\sigma \circ u \circ \sigma'$ means schedule $\sigma$ followed by $u$ followed by schedule $\sigma'$.

When $\sigma$ is a valid schedule for known set $W$, we define $M_W(\sigma)$ to be the peak memory taken by the schedule $\sigma' = w_1 \circ w_2 \circ \cdots \circ w_k \circ \sigma$, where we don't consider the memory for any $w_i$. Formally, suppose $\sigma' = u_1 u_2 \ldots u_{t'}$. As earlier, for $i \leq t'$, define

$$U_i = \{u_i\} \cup \text{in}(u_i) \cup \bigcup_{j > i} \{u' \in \text{in}(u_j) \mid \text{prev}(u', j) \leq i\}.$$

Then $M_W(\sigma) = \max_{i:u_i \notin W} s(U_i)$, where $s(U_i) = \sum_{u \in U_i} s(u)$. (We let $s(u)$ be the size of tensor output for node $u$, as before.) Notice that if we need to hold some $w_i$ in memory, we still must pay for it.

We define $L$ as the length of the schedule $\sigma$, as before, without including the $w_i$. Notice that $L(\sigma)$ makes sense even if $\sigma$ is not valid—it is still just the sum of the lengths of the operations in the schedule. So we can extend $L(\sigma)$ even for invalid sequences. We note that when $\sigma$ does not contain any nodes from $W$ that $L_w(\sigma) = L(\sigma)$ (where $\sigma$ may be valid for known set $W$ but not valid in general). Because of this, we will only consider $L(\sigma)$, regardless of the known set or validity of $\sigma$.

**Property 13.** *Let $W$ be a set of nodes, $x$ a node, and let $\sigma_1, \cdots, \sigma_k$ be schedules, with each $\sigma_i \circ x$ valid for known set $W$. Then $\sigma_1 \circ \cdots \circ \sigma_k \circ x$ is a valid schedule for known set $W$. Further,*

$$M_W(\sigma_1 \circ \cdots \circ \sigma_k \circ x) \leq \sum_i M_W(\sigma_i \circ x)$$

*We also have*

$$L(\sigma_1 \circ \cdots \circ \sigma_k \circ x) \leq \sum_i L(\sigma_i \circ x)$$

**Property 14.** *Let $X$ be a set of nodes $x_1 \prec x_2 \prec \cdots x_k$, $W$ a set of nodes, and $\sigma_1, \ldots, \sigma_k$ schedules such that for all $i$, the schedule $\sigma_i \circ x_i$ is valid for known set $W \cup X_{\prec x_i}$, where $X_{\prec x_i} = \{x \in X \mid x \prec x_i\}$. Then $\sigma_1 \circ x_1 \circ \sigma_2 \circ x_2 \cdots \circ \sigma_k \circ x_k$ is valid for known set $W$ and*

$$M_W(\sigma_1 \circ x_1 \circ \sigma_2 \circ x_2 \cdots \circ \sigma_k \circ x_k) \leq \max_i (M_{W \cup X_{\prec x_i}}(\sigma_i \circ x_i) + s(W \cup X_{\prec x_i}))$$

*We also have*

$$L(\sigma_1 \circ x_1 \cdots \circ \sigma_k \circ x_k) \leq \sum_i L(\sigma_i \circ x_i).$$

Note that this bound is a little sloppy. For the $k$th term, we could have simply used $M_{W \cup X_{\prec x_k}}(\sigma_i, x_k)$, avoiding the $s(W \cup X_{\prec x_k})$. It won't matter for our proofs.

We now introduce our key concept: interleaved schedules.

**Definition 15.** *Let $X$ be a set of nodes. An* interleaved schedule *on $X$ is a set of tuples $\mathcal{I} = \{\langle \sigma_1, x_1 \rangle, \langle \sigma_2, x_2, \rangle, \ldots \langle \sigma_k, x_k \rangle\}$ such that each $x_i \in X$ and each $\sigma_i$ is a schedule. We further require that for every $x \in X$, there is some tuple $\langle \sigma, x \rangle \in \mathcal{I}$.*

*Let $X_{\prec x} = \{x' \in X \mid x' \prec x\}$, and let $W$ be a set of nodes. We say $\mathcal{I}$ is* valid for known set $W$ *if for each $\langle \sigma, x \rangle \in \mathcal{I}$, the schedule $\sigma \circ x$ is valid for known set $W \cup X_{\prec x_i}$. If $W = \emptyset$, we say simply that $\mathcal{I}$ is* valid.

Suppose $\mathcal{I}$ in an interleaved schedule on $X$ with $\mathcal{I} = \bigcup_{i,j} \{\langle \sigma_{ij}, x_i \rangle\}$, where each $x_i$ is distinct and $x_1 \prec \cdots \prec x_k$ are the nodes in $X$. We define

$$M_W(\mathcal{I}) = \max_i \left( \sum_j M_{W \cup X_{\prec x_i}}(\sigma_{ij} \circ u_i) + s(W \cup X_{\prec x_i}\}) \right).$$

We also define

$$L(\mathcal{I}) = \sum_{i,j} L(\sigma_{ij} \circ u_i).$$

In addition, we need a few operations to combine interleaved schedules and convert them to [standard] schedules. Define

$$\text{Merge}(\mathcal{I}) = \bigcup_i \{\langle \sigma_i \circ x_i \rangle\},$$

where $\sigma_i = \sigma_{i1} \circ \sigma_{i2} \circ \cdots$ for each $i$. Within each $i$, the $\sigma_{ij}$ may be ordered arbitrarily. For concreteness, suppose they are ordered lexicographically.

Also define

$$\text{Flatten}(\mathcal{I}) = \sigma_1 \circ x_1 \circ \sigma_2 \circ x_2 \circ \cdots \circ \sigma_k \circ x_k.$$

Let $X' \subseteq X$, and let $i_1 < i_2 < \cdots < i_{k'}$ be indices so that $X' = \{x_{i_j}\}$, where $k' = |X'|$. Let $\tau_j = \sigma_{i_{j-1}+1} \circ x_{i_{j-1}+1} \circ \sigma_{i_{j-1}+2} \circ x_{i_{j-1}+2} \circ \cdots \sigma_{i_j}$, where $i_0 = 0$ for convenience. Define

$$\text{Condense}(X', \mathcal{I}) = \bigcup_j \{\langle \tau_j, x_{i_j} \rangle\}.$$

Notice that $\texttt{Condense}(X',\mathcal{I})$ is an interleaved schedule on $X'$. We also have that $\texttt{Condense}(X',\mathcal{I}) = \texttt{Condense}(X',\texttt{Merge}(\mathcal{I}))$ and $\texttt{Flatten}(\mathcal{I}) = \texttt{Flatten}(\texttt{Merge}(\mathcal{I})) = \texttt{Flatten}(\texttt{Condense}(X',\mathcal{I}))$.

**Property 16.** *Let $W$ be a set of nodes. Let $\mathcal{I}$ be an interleaved schedule on $X$ and let $\mathcal{I}'$ be an interleaved schedule on $X'$. Then*

$$M_W(\mathcal{I} \cup \mathcal{I}') \leq M_W(\mathcal{I}) + M_W(\mathcal{I}')$$

*and*

$$L(\mathcal{I} \cup \mathcal{I}') \leq L(\mathcal{I}) + L(\mathcal{I}').$$

*If $\mathcal{I}$ and $\mathcal{I}'$ are both valid for known set $W$, then $\mathcal{I} \cup \mathcal{I}'$ is valid for known set $W$.*

**Property 17.** *Let $W$ be a set of nodes. Let $\mathcal{I}$ be an interleaved schedule on $X$ and $\mathcal{I}'$ be an interleaved schedule on $X'$. Further, suppose $x \prec x'$ for all $x \in X, x' \in X'$. Then we have*

$$M_W(\mathcal{I} \cup \mathcal{I}') = \max\{M_W(\mathcal{I}), M_{W \cup X}(\mathcal{I}')\}.$$

*If $\mathcal{I}$ is valid for known set $W$ and $\mathcal{I}'$ is valid for known set $W \cup X$, then $\mathcal{I} \cup \mathcal{I}'$ is valid for known set $W$.*

**Property 18.** *Let $\mathcal{I}$ be an interleaved schedule on $X$ that is valid for known set $W$. Then $\texttt{Merge}(\mathcal{I})$ is an interleaved schedule on $X$ that is valid on known set $W$. Also, if $X' \subseteq X$, then $\texttt{Condense}(X',\mathcal{I})$ is an interleaved schedule on $X'$ that is valid on known set $W$.*

*Further, $M_W(\texttt{Merge}(\mathcal{I})) \leq M_W(\mathcal{I})$ and $L(\texttt{Merge}(\mathcal{I})) \leq L(\mathcal{I})$. Likewise, $M_W(\texttt{Condense}(X',\mathcal{I})) \leq M_W(\mathcal{I})$ and $L(\texttt{Condense}(X',\mathcal{I})) \leq L(\mathcal{I})$.*

**Property 19.** *Let $W$ be a set of nodes. If $\mathcal{I}$ is an interleaved schedule on $X$ that is valid for known set $W$, then $\texttt{Flatten}(\mathcal{I})$ is a valid schedule on known set $W$ that computes every node in $X$. Further,*

$$M_W(\texttt{Flatten}(\mathcal{I})) \leq M_W(\texttt{Merge}(\mathcal{I})) \leq M_W(\mathcal{I}),$$
$$L(\texttt{Flatten}(\mathcal{I})) \leq L(\texttt{Merge}(\mathcal{I})) \leq L(\mathcal{I}).$$

### A.3.1 Combining Interleaved Schedules with Path Decompositions

We need a few lemmas to better characterize interleaved schedules in the special cases we are considering. Let $\nu(\mathcal{X})$ be the set of nodes in $\mathcal{X}$.

**Lemma 20.** *Let $\mathcal{X} = (X_1, \ldots, X_b)$ be a path decomposition, and for some $i \in [b]$, let $\mathcal{X}' = (X_1, \ldots, X_{i-1})$. Let $\mathcal{I}$ be an interleaved schedule on $X$, and suppose that $\mathcal{I}$ is valid on known set $W \cup \overline{\nu(\mathcal{X}')}$, where $W \supseteq X_i$. Further, suppose that every node appearing in $\mathcal{I}$ is also in $\nu(\mathcal{X}')$. Then $\mathcal{I}$ is valid on known set $W \cup \overline{\nu(\mathcal{X})}$.*

*Proof.* Let $\langle \sigma, x \rangle \in \mathcal{I}$. We wish to show $\sigma \circ x$ is valid on known set $W \cup \overline{\nu(\mathcal{X})} \cup X_{\prec x}$. Take some $u \notin W \cup \overline{\nu(\mathcal{X})} \cup X_{\prec x}$ in $\sigma \circ x$, and consider some $v \in \text{in}(u)$. If $v \in W \cup X_{\prec x} \cup \overline{\nu(\mathcal{X})}$, then we are done. So suppose not. Then $v$ must appear in $\mathcal{X}$ but not in $X_{\prec x}$. (Since $v \prec u \prec x$, we see that $x$ cannot appear in $X$ either.)

If $v \in \nu(\mathcal{X}')$, then we are done, since $\sigma \circ x$ is valid on known set $W \cup X_{\prec x} \cup \overline{\nu(\mathcal{X}')}$. So suppose $v \notin \nu(\mathcal{X}')$ (but still $v \in \nu(\mathcal{X})$). There is an edge from $v$ to $u$, so $u$ and $v$ must appear in a bag together. Since $u \in \nu(\mathcal{X}')$ but $v \notin \nu(\mathcal{X}')$, either $u$ or $v$ must appear in $X_i \subseteq W$ by the betweenness property of path decompositions. That is a contradiction. So $\sigma \circ x$ is valid on known set $W \cup \overline{\nu(\mathcal{X})} \cup X_{\prec x}$.

The claim follows. $\qquad\square$

**Lemma 21.** *Let $\mathcal{X} = (X_1, \ldots, X_b)$ be a path decomposition, and for some $i \in [b]$, let $X = X_i$, let $\mathcal{X}^\ell = (X_1, \ldots, X_{i-1})$, and let $\mathcal{X}^r = (X_{i+1}, \ldots, X_b)$. Let $\mathcal{I}^\ell$ be an interleaved schedule on $X^\ell$, and let $\mathcal{I}^r$ be an interleaved schedule on $X^r$, and suppose $X^\ell \cap X^r = X$. Suppose that $\mathcal{I}^\ell$ is valid on known set $W \cup \overline{\nu(\mathcal{X}^\ell)}$ and $\mathcal{I}^r$ is valid on known set $W \cup \overline{\nu(\mathcal{X}^r)}$. Then $\texttt{Merge}(\mathcal{I}^\ell \cup \mathcal{I}^r)$ is valid on known set $W \cup \overline{\nu(\mathcal{X})}$.*

*Proof.* Choose $x \in X$, and let $\langle \sigma^\ell, x \rangle \in \texttt{Merge}(\mathcal{I}^\ell)$ and $\langle \sigma^r, x \rangle \in \texttt{Merge}(\mathcal{I}^r)$. We wish to show $\sigma^\ell \circ \sigma^r \circ x$ is valid on known set $W \cup \overline{\nu(\mathcal{X})} \cup X_{\prec x}$.

Take some $u$ in $\sigma \circ x$. If $u \in X$, without loss of generality, we can assume $u = x$. Consider $v \in \text{in}(x)$. If $v \in W \cup X_{\prec x} \cup \overline{\nu(\mathcal{X})}$, we are done. So suppose not. Then $v \notin X_{\prec x}$, which means $v \notin X$ since $v \prec x$. Further, $v \in \nu(\mathcal{X})$, which means either $v \in \nu(\mathcal{X}^\ell)$ or $v \in \nu(\mathcal{X}^r)$. Without loss of generality, suppose $v \in \nu(\mathcal{X}^\ell)$. Since $\sigma^\ell \circ x$ is valid on known set $W \cup X_{\prec x} \cup \overline{\nu(\mathcal{X}')}$, it must be the case that $v$ appears in $\sigma^\ell$. Hence, all vertices of $\text{in}(x)$ are either known or appear in the schedule before $x$.

Now, suppose $u \notin X$. Without loss of generality, say $u$ appears in $\sigma^\ell$. Consider some $v \in \text{in}(u)$. If $v$ appears in $W \cup X_{\prec x} \cup \overline{\nu(\mathcal{X})}$, then we are done. So suppose not. Then $v$ must appear in $\nu(\mathcal{X})$, but not in $X_{\prec x}$ (hence, not in $X$ since $v \prec x$). If $v \in \nu(\mathcal{X}^\ell)$, then, since $\sigma^\ell \circ x$ is valid for known set $W \cup X_{\prec x} \cup \overline{\nu(\mathcal{X}^\ell)}$, we're done. Otherwise, $v \in \nu(\mathcal{X}^r)$. But $v \in \text{in}(u)$, meaning that most $u$ and $v$ must appear in a bag together. By the betweenness property of path decompositions, that means either $u$ or $v$ must be in $X$. But that's a contradiction. Hence, all vertices of $\text{in}(u)$ are either known or appear in the schedule before $u$.

Hence, $\sigma^\ell \circ \sigma^r \circ x$ is valid on known set $W \cup \overline{\nu(\mathcal{X})}$, and the claim follows. $\qquad\square$

### A.3.2 Proofs for the Key Properties

*Proof of Property 13.* Let $\sigma' = \sigma_1 \circ \cdots \circ \sigma_k \circ x$. It is an immediate consequence of the definition that $\sigma'$ is valid on known set $W$. Consider some $u_i \notin W$ appearing in $\sigma'$ where the peak memory occurs. Either $u_i$ appears in $\sigma_\ell$ for some $\ell$, or $u_i$ is the last element in $\sigma'$ (i.e., node $x$). First, take the case that $u_i$ appears in $\sigma_\ell$ for some $\ell$. Recall the definition

$$U_i = \{u_i\} \cup \text{in}(u_i) \cup \bigcup_{j > i} \{u' \in \text{in}(u_j) \mid \text{prev}(u', j) \le i\},$$

and peak memory is $s(U_i)$. Let $U = \{u' \in \text{in}(u_j) \mid \text{prev}(u', j) \le i\}$, and let $U' = \{u' \in \text{in}(u_j) \mid \text{prev}(u', j) \le i$ such that $u_j$ appears in a sequence after $\sigma_\ell\}$. Notice that $M_W(\sigma_\ell \circ x) \ge s(U_i \setminus U')$, so we only need to bound $s(U')$.

For each $u \in U'$, there is some $u_j$ such that $u_j$ appears in $\sigma_{\ell'}$ with $\ell' > \ell$ and $u' \in \text{in}(u_j)$ with $\text{prev}(u', j) \le i$. Let $U^{\ell'}$ be the set of all such $u$. Then $s(U^{\ell'}) \le M_W(\sigma_{\ell'} \circ x)$. We have

$$s(U_i) = s(U_i \setminus U') + \sum_{\ell' > \ell} s(U^{\ell'}) \le M_W(\sigma_\ell \circ x) + \sum_{\ell' > \ell} M_W(\sigma_{\ell'} \circ x).$$

The claim follows for this case.

In the special case that $u_i$ is the last element in $\sigma'$, i.e., node $x$, say $x = u_{t'}$. Since $x$ appears last, our definition simplifies somewhat. We have $U_{t'} = \{x\} \cup \text{in}(x)$. Every node $u \in U_{t'}$ (for $u \notin x$) belongs to $\nu(\sigma_\ell)$ for some $\ell$. Let $U^\ell$ be the set of such $u$. We have

$$s(U_{t'}) = 1 + \sum_\ell s(U^\ell) \le \sum_\ell M_W(\sigma_\ell \circ x).$$

Our bound on $M_W$ follows.

To bound time, we simply note that every element appearing on the left-hand side of the inequality also appears on the right-hand side (with $x$ appearing multiple times). $\qquad\square$

*Proof of Property 14.* We induct on $k$. The case for $k = 1$ is trivially true. Consider general $k$. Let $\sigma = \sigma_1 \circ x_1 \cdots \circ \sigma_{k-1} \circ x_{k-1}$.

By induction, $\sigma$ is valid for known set $W \cup X_{\prec x_{k-1}}$. Since $\sigma_k \circ x_k$ is valid for known set $W \cup X_{\prec x_k}$, and all values of $X_{\prec x_k}$ appear in the schedule earlier than $\sigma_k$, we see $\sigma_1 \circ x_1 \cdots \circ \sigma_k \circ x_k$ is valid for known set $W$.

We now bound $M_W$. To do so, consider some segment of $\sigma$, say from $\sigma_\ell \circ x_\ell$. The set of nodes that must be held in memory for some future segment is at most $W \cup X_{\prec x_\ell}$. So if peak memory occurs during this segment, it is bounded by $M_{W \cup X_{\prec x_\ell}}(\sigma_\ell \circ x_\ell) + s(W \cup X_{\prec x_\ell})$. Since peak memory occurs in one of these segments, we have

$$M_W(\sigma_1 \circ x_1 \circ \sigma_2 \circ x_2 \cdots \circ \sigma_k \circ x_k) \le \max_i (M_{W \cup X_{\prec x_i}}(\sigma_i \circ x_i) + s(W \cup X_{\prec x_i}))$$

which is what we wanted.

Finally, the bound on time follows easily: every element on the left-hand side appears on the right-hand side. □

*Proof of Property 16.* Let $Z = X \cup X'$. Note that $\mathcal{I} \cup \mathcal{I}'$ is an interleaved schedule on $Z$. Also note that $Z_{\prec x} \supseteq X_{\prec x}$ and $Z_{\prec x} \supseteq X'_{\prec x}$ for any $x$.

We first show $\mathcal{I} \cup \mathcal{I}'$ is valid on known set $W$. For any $\langle \sigma, x \rangle \in \mathcal{I}$, we know $\sigma \circ x$ is valid for known set $W \cup X_{\prec x}$, hence it is also valid for known set $W \cup Z_{\prec x}$. Similarly, for any $\langle \sigma', x' \rangle \in \mathcal{I}'$, we see that $\sigma' \circ x'$ is a valid schedule for known set $W \cup Z_{\prec x}$. Hence, $\mathcal{I} \cup \mathcal{I}'$ is a valid interleaved schedule on known set $W$.

To see the memory bound, we have

$$
\begin{aligned}
M_W(\mathcal{I} \cup \mathcal{I}') &= \max_{x \in X \cup X'} \Big( \sum_{\sigma : \langle \sigma, x \rangle \in \mathcal{I} \cup \mathcal{I}'} M_{W \cup Z_{\prec x}}(\sigma \circ x) + s(W \cup Z_{\prec x}) \Big) \\
&\leq \max_{x \in X \cup X'} \Big( \sum_{\sigma : \langle \sigma, x \rangle \in \mathcal{I}} M_{W \cup Z_{\prec x}}(\sigma \circ x) + s(W \cup X_{\prec x}) \\
&\qquad + \sum_{\sigma : \langle \sigma, x \rangle \in \mathcal{I}'} M_{W \cup Z_{\prec x}}(\sigma \circ x) + s(W \cup X'_{\prec x}) \Big) \\
&\leq \max_{x \in X} \Big( \sum_{\sigma : \langle \sigma, x \rangle \in \mathcal{I}} M_{W \cup Z_{\prec x}}(\sigma \circ x) + s(W \cup X_{\prec x}) \Big) \\
&\qquad + \max_{x \in X'} \Big( \sum_{\sigma : \langle \sigma, x \rangle \in \mathcal{I}'} M_{W \cup Z_{\prec x}}(\sigma \circ x) + s(W \cup X'_{\prec x}) \Big) \\
&\leq \max_{x \in X} \Big( \sum_{\sigma : \langle \sigma, x \rangle \in \mathcal{I}} M_{W \cup X_{\prec x}}(\sigma \circ x) + s(W \cup X_{\prec x}) \Big) \\
&\qquad + \max_{x \in X'} \Big( \sum_{\sigma : \langle \sigma, x \rangle \in \mathcal{I}'} M_{W \cup X'_{\prec x}}(\sigma \circ x) + s(W \cup X'_{\prec x}) \Big) \\
&= M_W(\mathcal{I}) + M_W(\mathcal{I}').
\end{aligned}
$$

To bound the time, we have

$$
\begin{aligned}
L(\mathcal{I} \cup \mathcal{I}') &= \sum_{\langle \sigma, x \rangle \in \mathcal{I} \cup \mathcal{I}'} L(\sigma \circ x) \\
&\leq \sum_{\langle \sigma, x \rangle \in \mathcal{I}} L(\sigma \circ x) + \sum_{\langle \sigma, x \rangle \in \mathcal{I}'} L(\sigma \circ x) \\
&= L(\mathcal{I}) + L(\mathcal{I}').
\end{aligned}
$$

This completes the proof. □

*Proof of Property 17.* We first show $\mathcal{I} \cup \mathcal{I}'$ is valid for known set $W$. Let $Z = X \cup X'$, and note that $\mathcal{I} \cup \mathcal{I}'$ is an interleaved schedule on $Z$.

Let $\langle \sigma, x \rangle \in \mathcal{I} \cup \mathcal{I}'$. If $\langle \sigma, x \rangle \in \mathcal{I}$, then (since $\mathcal{I}$ is valid), we have $\sigma \circ x$ is valid on known set $W \cup X_{\prec x} \subseteq W \cup Z_{\prec x}$, which is what we wanted. On the other hand, if $\langle \sigma, x \rangle \in \mathcal{I}'$, then $\sigma \circ x$ is valid on known set $W \cup X \cup X'_{\prec x} \subseteq W \cup Z_{\prec x}$ since $x$ is topologically larger than all nodes in $X$. Hence, $\mathcal{I} \cup \mathcal{I}'$ is valid for known set $W$.

To bound the memory, first note that for $x \in X$, we have $Z_{\prec x} = X_{\prec x}$, and for $x \in X'$, we have $Z_{\prec x} = X \cup X'_{\prec x}$. Thus, we have

$$M_w(\mathcal{I} \cup \mathcal{I}') = \max_{x \in X \cup X'} \Big( \sum_{\sigma : \langle \sigma, x \rangle \in \mathcal{I} \cup \mathcal{I}'} M_{W \cup Z_{\prec x}}(\sigma \circ x) + s(W \cup Z_{\prec x}) \Big)$$

$$= \max \Big\{ \max_{x \in X} \Big( \sum_{\sigma : \langle \sigma, x \rangle \in \mathcal{I}} M_{W \cup X_{\prec x}}(\sigma \circ x) + s(W \cup X_{\prec x}) \Big),$$

$$\max_{x \in X'} \Big( \sum_{\sigma : \langle \sigma, x \rangle \in \mathcal{I}'} M_{W \cup X \cup X'_{\prec x}}(\sigma \circ x) + s(W \cup X \cup X'_{\prec x}) \Big) \Big\}$$

$$= \max \{ M_W(\mathcal{I}), M_{W \cup X}(\mathcal{I}') \}.$$

This completes the proof. $\qquad \square$

*Proof of Property 18.* If $\sigma \circ x$ is valid on known set $W'$ and $\sigma' \circ x$ is valid on known set $W'$, then clearly $\sigma \circ \sigma' \circ x$ is valid on known set $W'$. The proof for `Merge` follows from this observation and Property 13.

The reasoning for `Condense` is the same as in Property 14, and the proof follows from the definition of $M_W$ and $L$. $\qquad \square$

*Proof of Property 19.* We first show $\texttt{Flatten}(\mathcal{I})$ is a valid schedule. Let $u$ be some element in this schedule, and consider some $v \in \text{in}(u)$. Notice that $u$ corresponds to some element in $\mathcal{I}$, say it appears in $\sigma_\ell$ for some $\langle \sigma_\ell, x \rangle \in \mathcal{I}$. We know $\sigma_\ell \circ x$ is valid for known set $W \cup X_{\prec x}$. Hence, either $v \in W \cup X_{\prec x}$ or it appears in $\sigma_\ell$ earlier than $u$. The only potential problem is when $v$ does not appear earlier in $\sigma_\ell$ and $v \notin W$. In this case, $v \in X_{\prec x}$. But we know all items in $X_{\prec u} = X_{\prec x}$ have been produced earlier in $\texttt{Flatten}(\mathcal{I})$, so the schedule is valid.

To prove the memory bound, note that $\texttt{Flatten}(\mathcal{I}) = \texttt{Flatten}(\texttt{Merge}(\mathcal{I}))$, and by Property 13, $M_W(\texttt{Merge}(\mathcal{I})) \le M_W(\mathcal{I})$. So without loss of generality, we will assume that for each $x \in X$, there is exactly one $\sigma$ such that $\langle \sigma, x \rangle \in \mathcal{I}$.

But in this case, Property 14 shows directly that $M_W(\texttt{Flatten}(\mathcal{I})) \le M_W(\mathcal{I})$.

The time bound also follows directly from Property 13 and Property 14. $\qquad \square$

## A.4 Rematerialization Algorithm with Path Decomposition

In this section we present and analyze a rematerialization algorithm that works on path decompositions.

We present our main theorem.

**Theorem 22.** *Let $\mathcal{X} = \{X_1, \ldots, X_b\}$ be a path decomposition for a graph $G = (V, E)$, where $G$ is a directed acyclic graph with a directed path of length $n = |V|$ and whose max indegree is bounded by a constant. Let $u$ be the last node topologically in $G$, and let $\sigma = \texttt{PWRemat}(\mathcal{X}, u, \emptyset)$, as described in Algorithm 3. Then $\sigma$ is a valid schedule to compute $u$.*

*Furthermore, if the maximum bag size in $\mathcal{X}$, i.e., $\max_i \{|X_i|\}$, is bounded by $p$, then $L(\sigma) = O(2^p (p!)^2 b \log^{p-2} b)$ and $M(\sigma) = O((M_{\text{in}} + M_{\text{max}}) 2^p p \log^{p-2} b)$.*

We break the proof into two parts. The first shows correctness. The second bounds the time and space.

*Proof of correctness.* We prove two claims simultaneously using induction:

1. $\texttt{PWRemat}(\mathcal{X}, u, W)$ produces a schedule $\sigma$ that is valid schedule for known set $W \cup \overline{\nu(\mathcal{X})}$.

2. $\texttt{InterleavedSchedule}(X, \mathcal{X}, W)$ produces an interleaved schedule $\mathcal{I}$ on $X$ that is valid for known set $W \cup \overline{\nu(\mathcal{X})}$.

**Algorithm 3:** More Efficient Rematerialization via Path Decomposition.

---

**Function:** PWRemat($\mathcal{X}$, $u$, $W$):

  **Data:** $\mathcal{X} = (X_1, X_2, \ldots, X_b)$ a path decomposition, $u$ a node to compute, $W \subseteq V$ a subset of nodes whose value is known.

  **Result:** Returns a rematerialization schedule.

  **if** $u \in W \cup \overline{\nu(\mathcal{X})}$, *or every input to $u$ appearing in $\nu(\mathcal{X})$ is in $W$* **then**
      $\lfloor$ **return** *the schedule containing only $u$*

  Remove any nodes in $\mathcal{X}$ that are after $u$. Call this new decomposition $\mathcal{X}' = (X'_1, \ldots, X'_b)$;

  **if** $\mathcal{X}'$ *has pathwidth 1* **then**
      $\lfloor$ **return** *a valid schedule for $u$ for known set $W$ on nodes in $\nu(\mathcal{X}')$ with no rematerialization*

  Find $i$ such that $u \in X_i$;

  Compute $\mathcal{I}^\ell \leftarrow$ InterleavedSchedule($X'_i, \mathcal{X}^\ell, W$) where $\mathcal{X}^\ell = (X'_{i-1}, X'_{i-2}, \ldots, X'_1)$;

  Compute $\mathcal{I}^r \leftarrow$ InterleavedSchedule($X'_i, \mathcal{X}^r, W$) where $\mathcal{X}^r = (X'_{i+1}, X'_{i+2}, \ldots, X'_b)$;

  **return** Flatten($\mathcal{I}^\ell \cup \mathcal{I}^r$);

**Function:** InterleavedSchedule($X$, $\mathcal{X}$, $W$):

  **Data:** $X = \{x_1, x_2, \ldots, x_k\}$ a set of nodes, $\mathcal{X} = (X_1, X_2, \ldots, X_b)$ a path decomposition, $W \subseteq V$ a subset of nodes whose value is known.

  **Result:** Returns an interleaved schedule for $X$, i.e., a set of tuples $\langle \sigma, x \rangle$, where $x \in X$ and $\sigma$ is a schedule.

  **if** $X \subseteq W$ **then**
      $\lfloor$ **return** *an empty interleaved schedule on $X$, i.e. $\{\langle \emptyset, x \rangle \mid x \in X\}$*;

  Let $\mathcal{X}_{inner} = (X_1, X_2, \ldots, X_{b/2-1})$ and $\mathcal{X}_{outer} = (X_{b/2+1}, X_{b/2+2} \ldots, X_b)$;

  Initialize $\mathcal{I}_{inner} \leftarrow \emptyset$ and $W' \leftarrow W$, and let $X' = \{x \in X \mid x \prec last(X_{b/2})\}$;

  Let $u_1 \prec u_2 \prec \cdots \prec u_j$ be the nodes in $X' \cup X_{b/2}$;

  **for** $i = 1$ *to* $k$ **do**
      Assign $\sigma_i \leftarrow$ PWRemat($\mathcal{X}_{inner}, u, \{u_1, u_2, \ldots, u_{i-1}\}$) where $u$ is the node before $u_i$ in the spine for $G$;
      Add $\langle \sigma_1 \circ u_1 \circ \sigma_2 \circ u_2 \circ \cdots \sigma_i, u_i \rangle$ to $\mathcal{I}_{inner}$;

  Compute $\mathcal{I}_{outer} \leftarrow$ InterleavedSchedule($X_{b/2}, \mathcal{X}_{outer}, W$);

  Compute $\mathcal{I}_{lower} \leftarrow$ InterleavedSchedule($X \setminus X', \mathcal{X}_{inner}, W \cup X' \cup X_{b/2}$);

  **return** Condense($X, \mathcal{I}_{inner} \cup \mathcal{I}_{outer} \cup \mathcal{I}_{lower}$);

**Function:** Flatten($\mathcal{I}$):

  **Data:** An interleaved schedule, $\mathcal{I} = \bigcup_{i,j} \langle \sigma_{ij}, x_i \rangle$ with $x_1 \prec x_2 \prec \cdots$.

  **Result:** Returns a schedule.

  **return** $\sigma_1 \circ x_1 \circ \sigma_2 \circ x_2 \circ \cdots$, *where each* $\sigma_i = \sigma_{i1} \circ \sigma_{i2} \circ \sigma_{i3} \circ \cdots$;

**Function:** Condense ($X$, $\mathcal{I}$):

  **Data:** An interleaved schedule, $\mathcal{I} = \bigcup_{i,j} \langle \sigma_{ij}, x_i \rangle$ with $x_1 \prec x_2 \prec \ldots$.

  **Result:** Returns an interleaved schedule.

  **return** *an interleaved schedule $\mathcal{I}'$ on $X$ such that* Flatten($\mathcal{I}$) = Flatten($\mathcal{I}'$), *as defined in text.*

---

For the base case in Claim 1, we produce the schedule containing only $u$. Either $u \in W \cup \overline{\nu(\mathcal{X})}$ or all of its inputs are in $W \cup \overline{\nu(\mathcal{X})}$. In either case, $u$ (viewed as a schedule) is valid for known set $W \cup \overline{\nu(\mathcal{X})}$. In the case the $\mathcal{X}'$ has pathwidth 1, we again return a valid schedule for known set $W \cup \overline{\nu(\mathcal{X})}$.

For the base case in Claim 2, we produce an empty interleaved schedule on $X \subseteq W$, which is valid on known set $W$.

More generally, PWRemat($\mathcal{X}, u, W$) finds schedules $\mathcal{I}^\ell =$ InterleavedSchedule($X, \mathcal{X}^\ell, W$) and $\mathcal{I}^r =$ InterleavedSchedule($X, \mathcal{X}^r, W$) and returns Flatten($\mathcal{I}^\ell \cup \mathcal{I}^r$). By induction, $\mathcal{I}^\ell$ is

valid on known set $W \cup \overline{\nu(\mathcal{X}^\ell)}$ and $\mathcal{I}^r$ is valid on known set $W \cup \overline{\nu(\mathcal{X}^r)}$. Hence, by Lemma 21, $\texttt{Merge}(\mathcal{I}^\ell \cup \mathcal{I}^r)$ is valid on known set $W \cup \overline{\nu(\mathcal{X})}$. Hence, by Property 19, $\texttt{Flatten}(\mathcal{I}^\ell \cup \mathcal{I}^r) = \texttt{Flatten}(\texttt{Merge}(\mathcal{I}^\ell \cup \mathcal{I}^r))$ is valid on known set $W \cup \overline{\nu(\mathcal{X})}$, as we wanted.

Continuing, $\texttt{InterleavedSchedule}(X, \mathcal{X}, W)$ produces an interleaved schedule on $X$ consisting of the union of $\mathcal{I}_{inner}, \mathcal{I}_{outer}$, and $\mathcal{I}_{lower}$. We wish to show this union is an interleaved schedule that is valid on known set $W \cup \overline{\nu(\mathcal{X})}$.

We first consider $\mathcal{I}_{inner}$. We claim that after each iteration of the for loop, $\mathcal{I}_{inner}$ is valid on known set $\overline{\nu(\mathcal{X}_{inner})}$. To see this, consider tuple $\langle \sigma_1 \circ u_1 \circ \sigma_2 \circ u_2 \circ \cdots \circ \sigma_i \circ u_i \rangle$ added to $\mathcal{I}_{inner}$ on the $i$th iteration of the loop. In the base case, $i = 1$, we see that $\sigma_1$ is valid for known set $\overline{\nu(\mathcal{X}_{inner})}$ since it came from a call to $\texttt{PWRemat}(\mathcal{X}_{inner}, u, \emptyset)$. To see that adding $u_1$ does not change the validity, let $v \in \text{in}(u_1)$. If $v \notin \overline{\nu(\mathcal{X}_{inner})}$, then consider the spine from $v$ to $u$. That is, there are nodes $v = v_k, v_{k+1}, \ldots, v_\ell = u$ for which each $(v_j, v_{j+1})$ is an edge along the long spine in $G$. All of these nodes (other than $u$ itself) are ancestors of $u$, hence either all must appear in the valid schedule, or at least one must be known. (We assumed $v \notin \overline{\nu(\mathcal{X}_{inner})}$, so it cannot be known.) But say some ancestor of $u$, say $u'$, is known. Then $u' \in \overline{\nu(\mathcal{X}_{inner})}$, meaning it appears in a bag outside of $\mathcal{X}_{inner}$. By Lemma 9, this means there is some $v'$ on the path from $v$ to $u'$ such that $v$ is in $X'$ or $X_{b/2}$. But that is a contradiction, since $v' \prec u$ and $u$ is the first node in $X \cup X_{b/2}$. Hence, every $v \in \text{in}(u_1)$ appears in $\sigma_1$, meaning that $\sigma_1 \circ u_1$ is valid.

Continuing, again we wish to show the schedule $\sigma_1 \circ u_1 \circ \sigma_2 \circ u_2 \circ \cdots \circ \sigma_i \circ u_i$ is valid, for general $i$. By induction, it is not hard to see $\sigma_1 \circ u_1 \circ \sigma_2 \circ u_2 \circ \cdots \circ \sigma_i$ is valid. So we just need to argue that concatenating $u_i$ does not change validity. The proof is analogous to the base case. Let $v \in \text{in}(u_i)$ but $v \notin \overline{\nu(\mathcal{X}_{inner})} \cup \{u_1, u_2, \ldots, u_{i-1}\}$, and let $j$ be as small as possible so that $v \prec u_j$. We claim the part of the spine from $v$ to $u_j$ is contained in $\sigma_j$. To see this, all these nodes (other than $u_j$ itself) are ancestors of $u_j$, hence either must all appear in $\sigma_j$, or at least one must be known. Again, we assumed $v$ is not known. But suppose there is some other ancestor of $u$, say $u'$, on the spine is known. The $u' \in \overline{\nu(\mathcal{X}_{inner})} \cup \{u_1, \ldots, u_{i-1}\}$, meaning it appears in a bag outside of $\mathcal{X}_{inner}$; else, we did not choose the first $u_j$. By Lemma 9, this means there is some $v'$ on the path from $v$ to $u'$ such that $v$ is in $X'$ or $X_{b/2}$. But that is a contradiction, since $v' \prec u_j$ and $u_j$ is the first node in $X \cup X_{b/2}$ such that $v$ precedes it. Hence, every $v \in \text{in}(u_i)$ appears in $\sigma_j$, meaning that $\sigma_1 \circ u_1 \circ \sigma_2 \circ u_2 \circ \cdots \circ \sigma_i \circ u_i$ is valid.

Putting this together, we see that the $i$th tuple added to $\mathcal{I}_{inner}$ is valid on known set $\overline{\nu(\mathcal{X}_{inner})} \cup \{x_1, \ldots, x_{i-1}\}$. By Property 17, adding this tuple to $\mathcal{I}_{inner}$ produces a new interleaved schedule that is also valid on $\overline{\nu(\mathcal{X}_{inner})}$, hence valid on $W \cup \overline{\nu(\mathcal{X}_{inner})}$, as we wanted.

Next, $\mathcal{I}_{outer}$ is valid on known set $W \cup \overline{\nu(\mathcal{X}_{outer})}$ by induction. So we apply Lemma 21 to see that $\texttt{Merge}(\mathcal{I}_{inner} \cup \mathcal{I}_{outer})$ is valid on set $W \cup \overline{\nu(\mathcal{X})}$, as we wanted.

Finally, $\mathcal{I}_{lower}$ is valid on known set $W \cup X' \cup X_{b/2} \cup \overline{\nu(\mathcal{X}_{inner})}$ by induction. By Lemma 20, we have $\mathcal{I}_{lower}$ is valid on known set $W \cup X' \cup X_{b/2} \cup \overline{\nu(\mathcal{X})}$. Further, $\mathcal{I}_{lower}$ is an interleaved schedule on $X \setminus X'$, while $\texttt{Merge}(\mathcal{I}_{inner}, \mathcal{I}_{outer})$ is an interleaved schedule on $X' \cup X_{b/2}$. So by Property 17, $\texttt{Merge}(\mathcal{I}_{inner}, \mathcal{I}_{outer}) \cup \mathcal{I}_{lower}$ is valid on $W \cup \overline{\nu(\mathcal{X}_{inner})}$. The call to $\texttt{Condense}$ ensures that we return an interleaved schedule on $X$; notice that $\texttt{Condense}(\texttt{Merge}(\mathcal{I}_{inner}, \mathcal{I}_{outer}) \cup \mathcal{I}_{lower}) = \texttt{Condense}(\mathcal{I}_{inner} \cup \mathcal{I}_{outer} \cup \mathcal{I}_{lower})$, so we may omit the $\texttt{Merge}$.

This shows by induction that our two claims are true.

Given that (and in particular, the fact that the first claim is true), we see that when $\mathcal{X}$ is the path decomposition for $G$, $\texttt{PWRemat}(\mathcal{X}, u, \emptyset)$ produces a schedule $\sigma$ that is valid for known set $\emptyset \cup \overline{\nu(\mathcal{X})}$. But $\overline{\nu(\mathcal{X})} = \emptyset$, so $\sigma$ is a valid schedule to compute $u$. $\qquad \square$

Before finishing the proof, we will need the following lemma.

**Lemma 23.** *Consider Algorithm 3. Whenever $\texttt{InterleavedSchedule}()$ is called with a path decomposition of pathwidth $k$, the call to $\texttt{PWRemat}()$ operates on a path decomposition of pathwidth at most $k - 1$.*

*Proof.* This is a corollary of Lemma 11. Notice that it is clearer to think of operating on the induced interval graphs rather than a subgraph of $G$. □

*Proof of time and space for Theorem 22.* Let's now bound the time and memory taken by the schedule.

We first bound the maximum size of $W$, the known set, on any call. Notice that $W$ grows by at most $2p$ on any call, and the total recursive depth is $\log b$. Since we call PWRemat with $W = \emptyset$ initially, we see that it grows to size at most $2p \log b$. So $s(W) \le 2p M_{\max} \log b$ over all calls.

Next, a bit of notation. Let $T_p(b)$ be the time for a schedule $\sigma \circ u$ produced by a call to PWRemat$(\mathcal{X}, u, W)$ with path decomposition of length $b$ and maximum bag size $p$, and let $S_p(b)$ be its memory, for any $W$ such that $s(W) \le 2p M_{\max} \log b$. In particular, if PWRemat is called with known set $W$ (bounded as above) and $\mathcal{X}$ with maximum bag size of $p$ and it returns $\sigma$, then $S_p(b) = M_W(\sigma)$. Likewise, $T_p(b) = L(\sigma)$.

Similarly, let $T_p^{\mathcal{I}}(b)$ be the time for an interleaved schedule produced by InterleavedSchedule with path decomposition of length $b$ and maximum bag size $p$, and let $S_p^{\mathcal{I}}(b)$ be its memory, for any $W$ such that $s(W) \le 2p M_{\max} \log b$. Note that if $\mathcal{I}$ is an interleaved schedule returned by InterleavedSchedule with bounded known set $W$ and maximum bag size $p$, we have $S_p^{\mathcal{I}}(b) = M_W(\mathcal{I})$.

Throughout, we will assume that $T_p(b)$ is superlinear (otherwise, our schedule is linear in length). Hence, $T_p(b_1) + T_p(b_2) \le T_p(b_1 + b_2)$.

Looking at the call to PWRemat, we have

$$S_p(b) = m_W(\mathcal{I}^\ell \cup \mathcal{I}^r)$$
$$\le m_W(\mathcal{I}^\ell) + m_W(\mathcal{I}^r)$$
$$\le 2 S_p^{\mathcal{I}}(b).$$

Similarly,

$$T_p(b) \le T_p^{\mathcal{I}}(b_\ell) + T_p^{\mathcal{I}}(b_r) \le T_p^{\mathcal{I}}(b) \text{ by superlinearity.}$$

Here, $b_\ell$ and $b_r$ are the sizes of the left and right path decompositions.

Looking at the call to InterleavedSchedule, we first bound $M_W(\mathcal{I}_{inner})$. On the $i$th iteration of the loop creating $\mathcal{I}_{inner}$, we added the tuple $\langle \sigma_1 \circ u_1 \circ \sigma_2 \circ u_2 \circ \cdots \circ \sigma_i \circ u_i \rangle$. We have that each $\sigma_j$ is valid for known set $\{u_1, \ldots, u_{j-1}\}$, hence $\sigma_j \circ u_j$ is valid for known set $\{u_1, \ldots, u_{j-1}\} \cup \text{in}(u_j)$. Let $W = \bigcup_j \text{in}(u_j)$, and let $U = \{u_1, \ldots, u_k\}$. By Property 14, we can thus bound

$$M_W(\sigma_1 \circ u_1 \circ \sigma_2 \circ u_2 \circ \cdots \circ \sigma_i \circ u_i) \le \max_j \{M_{W \cup U_{\prec u_j}}(\sigma_j \circ u_j) + s(W \cup U_{\prec u_j})\}$$
$$\le \max_j \{M_{U_{\prec u_j}}(\sigma_j) + s(\text{in}(u_j)) + s(W \cup U_{\prec u_j})\}$$
$$\le S_{p-1}(b/2) + 2p M_{\max} + (2p+1) M_{in},$$

where the relation $M_{U_{\prec u_j}}(\sigma_j) \le S_{p-1}(b/2)$ follows from Lemma 11.

As we grew $\mathcal{I}_{inner}$ from $\emptyset$ to its final value, we repeatedly invoked Property 17. Each time we added some $\langle \sigma, u \rangle$ to the schedule; from above, $M_W(\sigma \circ u) \le S_{p-1}(b/2) + 2p M_{\max} + (2p+1) M_{in}$. So we can bound the value of $M_W(\mathcal{I}_{inner})$ by $S_{p-1}(b/2) + 2p M_{\max} + (2p+1) M_{in} \le S_{p-1}(b) + 3p(M_{\max} + M_{in})$.

Letting $W' = W \cup X' \cup X_{b/2}$, we have

$$S_p^{\mathcal{I}}(b) = m_W(\mathcal{I}_{inner} \cup \mathcal{I}_{outer} \cup \mathcal{I}_{lower})$$
$$\le \max\{M_W(\mathcal{I}_{inner}) + M_W(\mathcal{I}_{outer}), M_{W'}(\mathcal{I}_{lower})\}$$
$$\le \max\{S_{p-1}(b) + 3p(M_{\max} + M_{in}) + S_p^{\mathcal{I}}(b/2), S_p^{\mathcal{I}}(b/2)\}$$
$$= S_p^{\mathcal{I}}(b/2) + S_{p-1}(b) + 3p(M_{\max} + M_{in}).$$

Time is simpler. Note that $\mathcal{I}_{inner}$ consists of at most $2p$ tuples created from calls to PWRemat, where each tuple consists of at most $2p$ copies of schedules returned from PWRemat. So $L(\mathcal{I}_{inner}) \le$

$(2p)^2 L_{p-1}(b/2)$. We have

$$
\begin{aligned}
T_p^{\mathcal{I}}(b) &= L(\mathcal{I}_{inner} \cup \mathcal{I}_{outer} \cup \mathcal{I}_{lower}) \\
&\leq L(\mathcal{I}_{inner}) + L(\mathcal{I}_{outer}) + L(\mathcal{I}_{lower}) \\
&\leq (2p)^2 T_{p-1}(b/2) + T_p^{\mathcal{I}}(b/2) + T_p^{\mathcal{I}}(b/2) \\
&\leq 2T_p^{\mathcal{I}}(b/2) + 2p^2 T_{p-1}(b).
\end{aligned}
$$

Combining these inequalities, we have

$$
\begin{aligned}
S_p^{\mathcal{I}}(b) &\leq S_p^{\mathcal{I}}(b/2) + 2S_{p-1}^{\mathcal{I}}(b) + 3p(M_{\max} + M_{\mathrm{in}}) \\
T_p^{\mathcal{I}}(b) &\leq 2T_p^{\mathcal{I}}(b/2) + 2p^2 T_{p-1}^{\mathcal{I}}(b).
\end{aligned}
$$

Repeatedly replacing the first term on the right-hand side for each, we see

$$
\begin{aligned}
S_p^{\mathcal{I}}(b) &\leq S_p^{\mathcal{I}}(b/2^k) + 2k S_{p-1}^{\mathcal{I}}(b) + 3pk(M_{\max} + M_{\mathrm{in}}) \\
T_p^{\mathcal{I}}(b) &\leq 2^k T_p^{\mathcal{I}}(b/2^k) + 2p^2(T_{p-1}^{\mathcal{I}}(b) + 2T_{p-1}^{\mathcal{I}}(b/2) + \ldots 2^k T_{p-1}^{\mathcal{I}}(b/2^k)) \\
&\leq 2^k T_p^{\mathcal{I}}(b/2^k) + 2kp^2 T_{p-1}^{\mathcal{I}}(b).
\end{aligned}
$$

Hence,

$$
\begin{aligned}
S_p^{\mathcal{I}}(b) &\leq S_p^{\mathcal{I}}(1) + 2(\log b)S_{p-1}^{\mathcal{I}}(b) + 3p(\log b)(M_{\max} + M_{\mathrm{in}}) \\
&\leq 2(\log b)S_{p-1}^{\mathcal{I}}(b) + 4p(\log b)(M_{\max} + M_{\mathrm{in}}) \\
T_p^{\mathcal{I}}(b) &\leq bT_p^{\mathcal{I}}(1) + 2(\log b)p^2 T_{p-1}^{\mathcal{I}}(b) \\
&\leq 2(\log b)p^2 T_{p-1}^{\mathcal{I}}(b) + b.
\end{aligned}
$$

Unrolling each, we see

$$
\begin{aligned}
S_p^{\mathcal{I}}(b) &\leq 2^{p-2}(\log^{p-2} b)S_2^{\mathcal{I}}(b) + 4p(\log b)(M_{\max} + M_{\mathrm{in}})(1 + 2\log b + \ldots 2^{p-3}\log^{p-3} b) \\
&\leq 2^{p-2}(\log^{p-2} b)S_2^{\mathcal{I}}(b) + 2^p p(\log^{p-2} b)(M_{\max} + M_{\mathrm{in}}) \\
T_p^{\mathcal{I}}(b) &\leq 2^{p-2}(\log^{p-2} b)(p!)^2 T_{p-1}^{\mathcal{I}}(b) + b(1 + 2\log b + \ldots 2^{p-3}\log^{p-3} b) \\
&\leq 2^{p-2}(\log^{p-2} b)(p!)^2 T_{p-1}^{\mathcal{I}}(b) + 2^{p-2} b\log^{p-3} b.
\end{aligned}
$$

In the case when $p = 2$, we have a path, so $T_2^{\mathcal{I}}(b) = b$ and $S_2(b) = M_{\max} + M_{\mathrm{in}}$. So we have

$$
\begin{aligned}
S_p^{\mathcal{I}}(b) &\leq 2^{p+1} p(\log^{p-2} b)(M_{\max} + M_{\mathrm{in}}) \\
T_p^{\mathcal{I}}(b) &\leq 2^{p-1}(p!)^2 b(\log^{p-2} b).
\end{aligned}
$$

To complete the proof, note that $L(\sigma) = T_p(b) \leq T_p^{\mathcal{I}}(b)$ and $M(\sigma) = S_p(b) \leq 2S_p^{\mathcal{I}}(b)$. $\qquad\square$

To complete the argument for Theorem 7, we note that any path decomposition on $n$ nodes can be reduced to one of length $O(n)$ without increasing the pathwidth. The bounds follow.