[Reviews · NeurIPS 2019]

Reviewer 1



Though the paper is clearly written and presents an interesting approach for large model training. It also allows a GPU to train models that cannot fit into the GPU on-device memory. However, it completely ignores how fast a model can be trained after rematerialization. Given that it already takes a significant amount of time to train a large model on a single GPU and there are many alternatives to train large models without sacrificing too much on computation, it does not seem like the paper provides strong enough evidence over choosing an alternative approach. 1. The paper claims that the "feasibility" of training large models as the main motivation, but the memory itself is not the only bottleneck and the paper completely ignores the training speed side of the story. Training large models such as Bert on a single GPU already takes more than hundreds of days (e.g., Titan V), without specialized optimization. The evaluation results show that the reduced memory footprint is at the cost of 3-4 times increasing of the schedule length, which leads to at least 3-4 times increase of training time. The actual training time could be prohibitively slow, and sacrificing computation speed would further increase the already unbearable long training time and seems to be equally infeasible. It would be better that the paper reports the actual training time after rematerialization. 2. The main motivation of performing rematerialization is to make it possible to train a large model with limited memory, but the paper fails to discuss many alternatives that address the same problem. For example, existing DL frameworks such as TensorFlow and PyTorch support model parallelism, where it partitions the computation graph of a large model and uses aggregated device memory for training. On a single node, there are approaches that reduce memory consumption by reusing memory regions [1], or using unified memory that allows training to use both CPU and GPU memory [2, 3]. Given that, it is not clear the advantage of this work as compared with those existing work. 3. A large portion of the proposed technique is from rematerialization techniques in compilers, where tree decomposition is used to optimize register allocation. Algorithm 1 seems to be fairly generic and does not rely too much on any domain knowledge of neural networks. The technical contribution seems to be incremental. Minor: It seems to be possible that an increased schedule length can lead to much longer actual execution time. Is it possible that although the schedule length increases only 3-4 times, the new schedule may cause the computation to lose potential parallelism opportunities, which may increase the critical path and negatively impact the GPU utilization rate, leading to significantly longer execution time? It would be better to report the actual end-to-end training time. [1] K. Shirahata, Y. Tomita, and A. Ike. 2016. Memory reduction method for deep neural network training. In 2016 IEEE 26th International Workshop on Machine Learning for Signal Processing. [2] Chen Meng, Minmin Sun, Jun Yang, Minghui Qiu, and Yang Gu. 2017. Training deeper models by GPU memory optimization on TensorFlow. In Proc. of ML Systems Workshop in NIPS [3] M. Rhu, N. Gimelshein, J. Clemons, A. Zulfiqar, and S. W. Keckler. 2016. vDNN: Virtualized Deep Neural Networks for Scalable, Memory-Efficient Neural Network Design. ArXiv e-prints (Feb. 2016). arXiv:1602.08124 ===================== After author response ==================== After reading the author's response, I increased my score from 5 to 6 because although the technique has been applied to other domains, it has a good problem formalization and theoretical analysis of trading computation for memory consumption under the ML/DL context. It might not be a bad thing to release a paper which offers a different line of solving the memory consumption challenges in large-scale model training. That said, it would be better to consider the computation constraint while optimizing the memory usage.

Reviewer 2



This paper proposed a variant of gradient checkpointing algorithm that recursively applies checkpointing to derive a O(log n) memory cost for general computational graphs. First of all, I do believe that the general treatment of tree decomposition for computational memory optimization is valuable. Although some of that was also mentioned in the previous works (e.g. the tensorflow gradient checkpointing blog-post). It is helpful to have papers to formally summarize the algorithms. The authors uses “materialization” (comes from database systems) while most existing works uses gradient checkpointing (a terminology in AD). I think it would be helpful to clarify the relations of these terminologies to give readers a better context, as they are essentially the same thing. Given that recomputation trades computation for more memory, the authors should also list the additional computing cost besides the memory usage -- given that the recursive algorithm pays additional computing cost to save memory. I am not sure if we want to go as far as the recursive approach to pay the additional computing to save more memories -- given the sqrt(n) time cost might be good enough for many cases. Some discussion around computing cost vs memory would be helpful. Can the current algorithm be adapted to take a memory constraint into consideration to minimize the amount of compute needed for a memory constraint?

Reviewer 3



[Originality] This paper considers an interesting problem of rematerialization problem, with the motivation from the present situation of the complexity of neural networks and state-of-the-art machinery. [Quality & Clarity] * The general description using a DAG (in Section 2) is quite simple while I feel that it captures the motivation sufficiently. Given a DAG, where each vertex corresponds to computation and each edge corresponds data dependency, the objective is to find a short sequence of nodes with small peak memory, whose definition is natural. * The precise problem formulation and the proposed algorithm are not convincing, though Theorem 4 itself is technically sound. Instead of considering to optimize either length or peak memory of the schedule, the authors considered to provide an algorithm that computes a valid schedule whose length and schedule depends on treewidth. Since there is a tradeoff between the peak memory and schedule length, it is more natural to formulate an optimization problem of peak memory (or length) given a fixed length (or peak memory). E.g., Figure 3 shows that TwRemat runs 3 times as long as the other baselines; what if one wants to get a schedule that is twice as long as the baselines? Section 4.3 claims that we can interpolate TwRemat vs. NoRemat by changing the recursion limit, but it seems hard to find a schedule of desired length unless trying all the possible recursion limit. * Experimental evaluation is well described; the section reports the result of TwRemat and two baselines on several deep networks. It is effectively demonstrated that TwRemat can significantly reduce the peak memory at the price of schedule length, and the memory usage does not increase with the number of layers. It is also clearly presented that the treewidth of computation graphs of the real model is bounded. [Significance] Overall, I thought this is a reasonable paper. This work formulates a new rematerialization problem for complex neural networks, develops an algorithm that works very well for bounded-treewidth cases, and verifies the effectiveness by experimental evaluations using real deep networks.

[Author Response · NeurIPS 2019]

We thank the reviewers for their thoughtful comments. We will address their comments in the revised version of the paper. We believe our contributions will be interesting to the ML community for several reasons:

- General, ***automatic*** methods to reduce memory consumption for large deep learning models are critical. To our knowledge, techniques up to this point have been mostly focused on special case models. In practice, getting extremely large models to run on GPUs or TPUs is a time-consuming task, requiring expert knowledge of the problem.

- As Reviewer 1 pointed out, it is not always clear that the increase in running time is worth the savings in memory. But consider the case of training BERT on a TPU pod, which takes around 4 days. If that model doubled in size, it could take weeks of engineering work to get it to run. Our algorithm offers the ability to greatly reduce the memory with no human intervention, at the cost of an extra week of computation; our personal experience suggests this is a tradeoff that many would happily make. Of course, there are cases on both sides of this tradeoff, but having a reasonable starting point is valuable in itself.

- We provide a formalization of the problem with rigorous guarantees. As several reviewers noted, this is a useful contribution. In fact, while using treewidth to improve rematerialization has been suggested in an informal setting before (the tensorflow gradient checkpointing blog post), we show later in this rebuttal that the primary theoretical arguments therein are incorrect. In addition, we feel that our formalization will make the problem more attractive to the algorithmic community within NeurIPS, which would help garner additional insights. (For example, the fact that low pathwidth graphs have better theoretical guarantees is quite surprising, even if it doesn't immediately lend itself to practical algorithms.)

We now address a few of the specific reviewer concerns.

**Minimizing schedule length with memory constraint.** We have heuristics based on the TwRemat algorithm that, for a given memory limit, work by stopping the recursion early when the current set of nodes to compute can be scheduled without rematerialization under the memory limit. While minimizing the execution time of the schedule given a hard constraint on memory is an ideal version of our problem, we felt that including those heuristics would muddy the clarity of our results. However, in the revised version of this paper we will include a more thorough discussion of this. From a theoretical viewpoint, no algorithm with runtime sub-exponential in treewidth can yield a provable approximation guarantee on the schedule length (under common complexity assumptions). We will also add a discussion regarding this hardness result in the revised version.

**Comparisons with model parallelism.** While it is true that model parallelism is also a way to reduce the memory requirements of training, rematerialization and model parallelism are orthogonal techniques. One can simultaneously apply the two for even greater memory savings; hence our work is complementary to model parallelism. In addition, our work makes rematerialization easy to automate, whereas most model parallelism is achieved by careful hand-crafting. Since our rematerialization algorithms operate automatically on all networks, the neural network designer can obtain the benefits of memory reduction for "free" without any specialized optimization.

**Addressing the blog post.** Unfortunately the algorithm suggested by the TensorFlow checkpointing blog post is incorrect. That post draws on Courcelle's theorem (namely, every graph property definable in the monadic second-order logic of graphs is linear-time decidable on bounded treewidth graphs) to characterize the problems that become tractable for graphs of small treewidth. However, rematerialization is known to be PSPACE-complete [1]. Writing down a monadic second-order logic formula for rematerialization would imply that the problem is in PH (the polynomial hierarchy), and hence we would arrive at the surprising result that PSPACE collapses to PH. Thus, it is unlikely that we could apply Courcelle's theorem to rematerialization.

**Comparisons with previous work.** We thank the reviewers for pointing out the references that we missed and will be sure to include a comparison with those approaches in the revised version. We feel that it's more accurate to avoid the term "gradient checkpointing" here, because our technique can be applied even to inference graphs, but we will definitely include a discussion of terms.

# References

[1] John R Gilbert, Thomas Lengauer, and Robert Endre Tarjan. The pebbling problem is complete in polynomial space. In *Proceedings of the 11th annual ACM Symposium on Theory of Computing*, pages 237–248. ACM, 1979.


[Meta-Review · NeurIPS 2019]

The paper present a new approach for rematerialization that can trade off memory and computation based on tree decomposition of the computation graph. In contrast to previous approaches in ML / AD that addressed this optimization in the limited context of gradient calculation, the current paper address it as a general scheduling problem of a computational graph. The complexity of the algorithm and the overhead of recomputation is bound by the tree width of the graph. Given the novelty of the approach I would like to accept this paper despite the weakness of the paper especially the lack of consideration of the actual computation cost. More detailed analysis and better way to control the trade-off of (actual) computation and memory would make the paper much stronger.